# Enzymatic Formation of Polyaniline, Polypyrrole, and Polythiophene Nanoparticles with Embedded Glucose Oxidase

**DOI:** 10.3390/nano9050806

**Published:** 2019-05-27

**Authors:** Natalija German, Anton Popov, Almira Ramanaviciene, Arunas Ramanavicius

**Affiliations:** 1NanoTechnas – Center of Nanotechnology and Materials Science, Faculty of Chemistry and Geosciences, Vilnius University, Naugarduko 24, LT-03225 Vilnius, Lithuania; natalija.german@imcentras.lt (N.G.); anton.popov@chgf.vu.lt (A.P.); almira.ramanaviciene@chf.vu.lt (A.R.); 2Department of Immunology, State Research Institute Centre for Innovative Medicine, Santariskiu 5, LT-08406 Vilnius, Lithuania; 3Department of Physical Chemistry, Faculty of Chemistry and Geosciences, Vilnius University, Naugarduko 24, LT-03225 Vilnius, Lithuania; 4Division of Materials Science and Electronics, State Scientific Research Institute Center for Physical Sciences and Technology, Savanorių ave. 231, LT-02300 Vilnius, Lithuania

**Keywords:** glucose oxidase, polyaniline, polypyrrole, polythiophene, nanoparticles, polymerization, spectrophotometry

## Abstract

Polyaniline (PANI), polypyrrole (Ppy), and polythiophene (PTh) composite nanoparticles with embedded glucose oxidase (GOx) were formed by enzymatic polymerization of corresponding monomers (aniline, pyrrole, and thiophene). The influence of monomers concentration, the pH of solution, and the ratio of enzyme/substrate on the formation of PANI/GOx, Ppy/GOx, and PTh/GOx composite nanoparticles were spectrophotometrically investigated. The highest formation rate of PANI-, Ppy-, and PTh-based nanoparticles with embedded GOx was observed in the sodium acetate buffer solution, pH 6.0. The increase of optical absorbance at *λ*_max_ = 440 nm, *λ*_max_ = 460 nm, and *λ*_max_ = 450 nm was exploited for the monitoring of PANI/GOx, Ppy/GOx and PTh/GOx formation, respectively. It was determined that the highest polymerization rate of PANI/GOx, Ppy/GOx, and PTh/GOx composite nanoparticles was achieved in solution containing 0.75 mg mL^−1^ of GOx and 0.05 mol L^−1^ of glucose. The influence of the enzymatic polymerization duration on the formation of PANI/GOx and Ppy/GOx composite nanoparticles was spectrophotometrically investigated. The most optimal duration for the enzymatic synthesis of PANI/GOx and Ppy/GOx composite nanoparticles was in the range of 48–96 h. It was determined that the diameter of formed PANI/GOx and Ppy/GOx composite nanoparticles depends on the duration of polymerization using dynamic light scattering technique (DLS), and it was in the range of 41–167 nm and 65–122 nm, when polymerization lasted from 16 to 120 h.

## 1. Introduction

Conducting polymers (CPs) such as polyaniline (PANI), polypyrrole (Ppy), and polythiothene (PTh) are very attractive due to the number of technological advantages [1,2,3,4]. The electrical conductivity of PTh, Ppy, and PANI reaches 10, 1 × 10^2^, and 1 × 10^3^ S cm^−1^, respectively [1,5,6,7,8]. Therefore, nanocomposites based on these CPs are especially interesting from the applicability-related point of view [9,10,11,12,13], because these CPs are nontoxic and stable in aggressive chemical environments. In addition, CPs possess unique electrical, mechanical, and charge transfer properties, which are also very attractive for many practical applications [1,14,15,16].

Most of the CPs are insoluble in the common organic solvents, which are explained by favourable interactions between *π*-systems of proximal polymeric chains [7,17]. Such interactions are stronger than between the polymer and the solvent; therefore, CPs are forming nanoparticles and/or larger aggregates [17]. PANI, Ppy, and PTh can be dependently doped by corresponding ions on the pH value of polymerization bulk solution during oxidation or reduction processes [1,5,6,18,19,20]. Novel strategies for the enzymatic synthesis of PANI [21,22], Ppy [23,24,25,26], and PTh [27] were developed during the last decade. In this context, the enzymatic polymerization seems to be very attractive method, which is suitable for the formation of CPs, because it is carried out in neutral aqueous solutions containing redox enzyme [21,22,24,25,27].

PANI is commonly used in the fabrication of electrochemical biosensors, due to controllable conductivity, good processability, controllable degree of protonation, charge transfer capability, good biocompatibility, and relatively good environmental stability [8,20,28]. PANI exists in three well-defined oxidation states: leucoemeraldine, emeraldine, and pernigraniline [1,19]. The oxidation/reduction of PANI depend on the pH value. PANI can also be in the form of bases or protonated salts, which are characterized by different colours (yellow, green, blue, or violet) and conductivities [19]. The amine and imine groups of PANI can be protonated or deprotonated depending on the pH values [19]. Conducting (4–10 S cm^−1^) PANI is produced during chemical polymerization in the solution of high acidity (pH < 2.5). The morphology of formed PANI composite depends on the pH value [5,8,19]. In the pH range of 2.5 to 4.6, a considerable part of aniline is protonated, but the imine groups in polymeric chains still remain unprotonated. Therefore, oxidative processes are attenuated and the rate of proton release is decreased [6]. PANI presents in the form of neutral molecules and it can be more easily oxidized than anilinium cation, which dominates at higher acidity (pH < 1.0) under the conditions of low acidity (pH 4.6) [5,28]. Oxidized terminal imine groups of PANI replace hydrogen atoms in benzene rings of the neutral aniline molecules, then the polymerization reaction is occurring at pH > 4.6. Furthermore, PANI is not formed in regular polymer structure, but in the form of oligomers of different structures, based on *orto*- and *para*-coupled units with cyclic phenazine fragments [6,19].

One of the most widely investigated CPs is Ppy, which is characterized by a relatively stable electrical conductivity and it can be formed by electrochemical synthesis at low oxidation potentials and in solutions of neutral pH [7,18,20,24,26]. Ppy in aqueous solution can be synthesized by various methods and it can be used in the design of biosensors that are based on immobilized enzymes, antibodies, or DNA [26,29,30,31]. Black oligomers of pyrrole start to form solid particles and then the polymerization is continuing on the surface of such polymeric particles [18]. Some cross-linked Ppy-based structures are formed after a certain time of polymerization reaction [18]. Generated by the oxidation of the pyrrole cation radicals are deprotonated and neutral radicals are formed during chemical polymerization [18].

Conducting polymer PTh has a great potential for the application in the fields of organic electronics, photovoltaic devices, solar cells, gas, and biological sensors [16,32,33]. High environmental stability, electrical conductivity, useful redox, and optical properties characterize PTh [13,15,16,26,33,34]. The molecule of PTh has *π*-*π* conjugated sp^2^ hybridized backbone [32]. Electrochemical [16,35], chemical [16,34], and enzymatic [27] polymerization based methods are used for PTh formation. The polymerization of thiophene is relatively easy and it proceeds via electrochemical polymerization, chemical oxidative polymerization, and Suzuki polycondensation based routes [16,32]. During the polymerization, firstly, the thiophene monomer is oxidized to a cation and was combined with another ionized or electrostatically neutral monomer’s molecule [16]. The dimer is further connected with another monomer by the same route [16]. Oxidized and reduced forms of PTh are red/brown and blue coloured, respectively [34]. The structure of PTh is mostly regular. The formation of soluble PTh is still a challenge [12,13], because PTh is usually formed in insoluble form, even if polymerization is performed in organic solutions [32,34,35,36]. Depending on polymerization conditions [12,32], PTh can be synthesized in form of spherical nanoparticles [12,13,36], thin films [34,35,37], or nanocomposites [34].

Enzymatic polymerization has been proven by PANI formation while using enzymes (glucose oxidase [21], horseradish peroxidase [22,38], and laccase [38], which catalyse the generation of respective aniline free radicals [19]. Peroxidases contain a heme cofactor in their active sites, or alternatively redox-active cysteine or seleno-cysteine residues [38]. The action of peroxidases mostly is based (i) on a two electron transfer step or (ii) on two clearly observable subsequent single electron transfer steps, which enable the reduction of peroxides to water or alcohols [38]. The most optimal substrate is hydrogen peroxide for many peroxidases, which is produced in some biological oxidation reactions [22,38]. The hydrogen peroxide acts as an oxidizer in oxidative-polymerization reactions, which are used for formation of conducting polymers, which is generating cation-radicals that tends to form polymeric structures [18,19,27].

Glucose oxidase (GOx) was used for the enzymatic formation of PANI, Ppy, and PTh composite nanoparticles in this research, with embedded GOx (PANI/GOx, Ppy/GOx, and PTh/GOx), respectively. The effect of pH, the ratio of enzyme/substrate on the formation of these composite nanoparticles and the influence of polymerization duration on the formation of PANI/GOx, Ppy/GOx were investigated by UV/Vis spectrophotometry. The diameter of the PANI/GOx, Ppy/GOx, and PTh/GOx composite nanoparticles was evaluated while using the dynamic light scattering (DLS) technique.

## 2. Materials and Methods

### 2.1. Materials

Glucose oxidase (EC 1.1.3.4, type VII, from *Aspergillus niger*, 201 units mg^−1^ protein) and D-(+)-glucose were purchased from Fluka (Buchs, Switzerland) and Carl Roth GmbH + Co.KG (Karlsruhe, Germany). Glucose solution was allowed to mutarotate for 24 h before all of the investigations, and the equilibrium between α and β optical isomers was established during this time. All other chemicals that were used in present study were either analytically pure or of highest quality. All of the solutions were prepared using deionized water. The solution of sodium acetate (SA) buffer (0.05 mol L^−1^ CH_3_COONa·3H_2_O) with 0.1 mol L^−1^ KCl was prepared by the mixing of sodium acetate trihydrate and potassium chloride, which both were purchased from Reanal (Budapest, Hungary) and Lachema (Neratovice, Czech Republic). Alfa alumina powder (Al_2_O_3_, grain diameter 0.3 micron, Type N) was purchased from Electron Microscopy Sciences (Hatfield, MA, USA). Aniline, sodium hydroxide (NaOH), and thiophene were purchased from Merck KGaA (Darmstadt, Germany), pyrrole — from Acros Organics (New Jersey, NJ, USA) and hydrochloric acid (HCl) — from Sigma-Aldrich (Saint Louis, MO, USA). The polymers were filtered before measurements through 5 cm column that was filled by Al_2_O_3_ powder to remove the coloured components. All of the solutions were stored between measurements at +4 °C.

### 2.2. The Formation and Separation of PANI/GOx, Ppy/GOx and PTh/GOx Composite Nanoparticles

PANI, Ppy, and PTh composite nanoparticles with embedded GOx were synthesized at room temperature (+20 ± 2 °C) in darkness while using polymerization bulk solution containing 0.05 mol L^−1^ of glucose, 0.75 mg mL^−1^ of GOx, and 0.50 mol L^−1^ of aniline, pyrrole, or thiophene. The enzymatically formed polymeric nanoparticles were separated from the polymerization bulk solution by the centrifugation (6 min, 16.1 × 10^3^ g) with IEC CL31R Multispeed centrifuge (ZI Aze Bellitourne, France). Subsequently, PANI/GOx, Ppy/GOx, and PTh/GOx composite nanoparticles were washed two times with 0.05 mol L^−1^ SA buffer, pH 6.0, and they were collected by the centrifugation. Separated and washed polymeric nanoparticles with embedded GOx were resuspended in SA buffer, pH 6.0, and used for further investigations. All of the experiments were performed at +20 ± 2 °C. Figure 1 illustrates the main steps of enzymatic polymerization and the formation of polymeric nanoparticles.

### 2.3. The Optimization of PANI/GOx, Ppy/GOx and PTh/GOx Composite Nanoparticles Formation Conditions

The efficiency of polymeric nanoparticles formation depends on the pH of solution and the duration of enzymatic polymerization. The optimal pH value for the formation of PANI/GOx, Ppy/GOx and PTh/GOx composite nanoparticles was studied in pH range from pH 1.0 to 12. Polymerization bulk solution containing 0.05 mol L^−1^ of glucose, 0.50 mol L^−1^ of polymerizable monomers (aniline, pyrrole, or thiophene), and 0.75 mg mL^−1^ of GOx was used for 48 h lasting polymerization. Deionized water, 0.1, 0.01, or 0.001 mol L^−1^ HCl solutions, 0.05 mol L^−1^ SA buffer, pH 6.0, and 1 × 10^−6^, 1 × 10^−4^ or 0.01 mol L^−1^ NaOH solutions were used to prepare solutions in pH range between 1.0 and 12. During 48 h lasting polymerization, several concentrations of glucose (were 0.015, 0.02, 0.03, 0.04, 0.05, 0.06 and 0.1 mol L^−1^) were tested in the 0.05 mol L^−1^ SA buffer, pH 6.0, containing 0.75 mg mL^−1^ of GOx and 0.50 mol L^−1^ of aniline, pyrrole, or thiophene. The optimal duration of polymerization was determined by varying the duration of polymerization from 5 to 288 h in 0.05 mol L^−1^ SA buffer, pH 6.0, with 0.05 mol L^−1^ of glucose, 0.50 mol L^−1^ of aniline or pyrrole, and 0.75 mg mL^−1^ of GOx. A UV/Vis spectrophotometer was used to perform the investigations.

### 2.4. The Monitoring of PANI/GOx, Ppy/GOx, and PTh/GOx Formation by UV/Vis Spectrophotometry

The UV/Vis spectrophotometer Lambda 25 (Shelton, CT, USA), which operates in 230–1000 nm wavelength range, was used for the registration of optical absorbance spectra of the formed PANI/GOx, Ppy/GOx, and PTh/GOx composite nanoparticles colloidal solution. The measurements were performed at room temperature in plastic disposable cuvettes of 1 cm optical path length. The optical absorbance of PANI/GOx, Ppy/GOx, and PTh/GOx solutions was registered after the polymerization reaction. UV/Vis measurements were evaluated and presented using SigmaPlot software 12.5 (San Jose, CA, USA).

### 2.5. The Evaluation of PANI/GOx, Ppy/GOx, and PTh/GOx Composite Nanoparticles Formation by DLS

The influence of solution’s pH on the diameter and zeta-potential of PANI/GOx, Ppy/GOx, and PTh/GOx composite nanoparticles was studied in pH range from 2.0 to 8.0 at +20 ± 2 °C. In this experiment, 0.01 or 0.001 mol L^−1^ HCl, 0.05 mol L^−1^ SA buffer, pH 6.0, and 1 × 10^−6^ mol L^−1^ NaOH were used to prepare solutions in pH range between 2.0 and 8.0. Polymerization solutions containing 0.05 mol L^−1^ of glucose, 0.50 mol L^−1^ of aniline, pyrrole, or thiophene, and 0.75 mg mL^−1^ of GOx were monitored during 15–120 h lasting polymerization.

Polymeric nanoparticles after enzymatic synthesis were separated, washed two times with 0.05 mol L^−1^ SA buffer, pH 6.0, and collected by the centrifugation (6 min, 16.1 × 10^3^ g), as it was described in chapter 2.2. The diameter and zeta-potential of PANI/GOx, Ppy/GOx, and PTh/GOx composite nanoparticles were evaluated by DLS and microelectrophoresis, while using the Zetasizer Nano ZS from Malvern (Herrenberg, Germany) equipped with a 633 nm He-Ne laser. The obtained data was analysed with Dispersion Technology Software version 6.01 from Malvern. DLS and microelectrophoresis investigations were evaluated and visualized by SigmaPlot software 12.5.

### 2.6. Imaging of PANI/GOx and Ppy/GOx Composite Nanoparticles by field Emission Scanning Microscope FE-SEM

The enzymatic polymerization of the PANI/GOx and Ppy/GOx composite nanoparticles was proceeded in darkness at room temperature within 24 h. The procedure of the formation, separation, washing, and collection of nanoparticles was similarly performed, as it was described in chapter 2.2. To reduce the effect of dissolved salts, colloidal solutions of composite nanoparticles were diluted in 50 μL of deionized water and homogenized using an ultrasound Bandelin Sonorex RK 31 H (Berlin, Germany). Afterwards, 6 μL of nanoparticle solution was placed on the surface of graphite rod and dried, this procedure was repeated eight times, and then the surface was characterized by Hitachi SU-70 field emission scanning microscope equipped by energy dispersive X-ray spectrometry EDS and electron backscatter diffraction (EBSD) modules from Hitachi (Dublin, Ireland). Turbosputer was applied for the preparation of synthesized CPs for SEM-based imaging.

## 3. Results and Discussion

### 3.1. The Influence of Polymerization Solution pH on PANI/GOx, Ppy/GOx and PTh/GOx Composite Nanoparticles Formation

The large number of research papers is dealing with the chemical and electrochemical polymerization of aniline, pyrrole, and thiophene, but only few researches are dealing with the enzymatic formation of these CPs. The enzymatic formation of these CPs can be based on the application of four major compounds: corresponding monomer, glucose oxidase, glucose, and oxygen, which is naturally dissolved in used solution (Figure 1).

GOx generates hydrogen peroxide and gluconolactone in the presence of glucose and dissolved oxygen, which is hydrolyzed to gluconic acid in an aqueous solution. Hydrogen peroxide as a strong oxidizer generates radical cation of pyrrole, aniline [18,19,25], and, in such a way, it initiates the polymerization reaction in acidic environment that is created by gluconic acid [27]. These free radical cations undergo coupling and, in this way, oligomers and polymeric structures are formed [18,19]. The diffusional permeability of formed polymeric structures is sufficient; therefore, a decent substrate and product mobility towards/outwards the enzyme, which is encapsulated within the formed polymer layer, is retained [27].

PANI and Ppy are very often used for the immobilization of biomolecules [1]. Neutral pH values are required for the enzymatic synthesis of PANI- and Ppy-based composites [21,24,39]. However, good polymerization efficiency at neutral pHs is difficult to achieve; in addition, PANI and Ppy usually are better conducting and are more electroactive only in their protonated form, which is formed at low pH [1]. The concentration of the reactant and the velocity of oxidation process both increase at low values of pH [36]. At pH > 4.6, both aniline and pyrrole monomers and corresponding polymer chains are not protonated and, in such case, only neutral nitrogen-containing species are involved in PANI or Ppy formation reactions which are characterized by low oxidation potential [6]. Therefore, the pH value of the polymerization solution is a very important parameter for the formation of PANI or Ppy [18].

The enzymatic formation of PANI-, Ppy-, and PTh-based composite nanoparticles with embedded glucose oxidase was monitored by UV/Vis spectrophotometry in the pH interval from 1.0 to 12. During the first few minutes, the polymerization bulk solutions containing aniline, pyrrole, or thiophene were colourless and any absorbance peaks were observed in the visible part of spectra. The formation of polymeric nanoparticles (PANI/GOx, Ppy/GOx, or PTh/GOx, respectively) was observed after 24 h of polymerization. In the case of the formation of PANI/GOx and Ppy/GOx, the solutions become coloured yellow-brown and black-grey, respectively. It was later observed that the intensity of colour and the value of optical absorbance were dependent on the pH value of the polymerization solution. Preliminary spectrophotometric investigations showed that the enzymatic polymerization of aniline [21], pyrrole [24,25] and thiophene [12,15] was followed by the formation of peaks with optical absorbance maximum (*λ*_max_) at 440 nm, 460 nm, and 450 nm, respectively (Figure 2A, Figure 3A and Figure 4A). These *λ*_max_ corresponds to the formation of aniline, pyrrole and thiophene oligomers. The optical absorbance peaks at *λ*_max_ = 260 nm were registered in all cases; this absorbance can be explained by a presence of protein—glucose oxidase—in the polymerization solutions, because amino acid phenylalanine optical absorbance peak is at *λ*_max_ = 260 nm.

PANI/GOx colloidal solution after 48 h lasting enzymatic polymerization was characterised by one for PANI characteristic optical absorbance peak with maximum *λ*_max_ = 440 nm (Figure 2A), which is in agreement with another investigations related to synthesis of PANI [8,11,28]. Some of the authors reported the optical absorbance peak at *λ*_max_ = 360 nm, which is attributed to emeraldine salt, corresponding to *π*-*π** transition of PANI benzene ring delocalized on nitrogen atoms and related to the formation of cyclic aniline dimer 5,10-dihydrophenazine [8,11,28]. It was reported that optical absorbance peaks at *λ*_max_ =360 nm and *λ*_max_ = 420 nm are able to combine into a flat or distorted single peak with a new maximum being formed between these initial two peaks [28]. The oxidation of 5,10-dihydrophenazine generates phenazylium radical-cation, which absorbs at *λ*_max_ = 440 nm. Such a phenazylium radical-cation is usually insoluble in the most organic solvents and in aqueous medium of low pH value [6]. All of the PANI/GOx composite nanoparticles that formed in polymerization bulk solution in the pH range from 5.0 to 8.0 were formed in the emeraldine oxidation form, which has been confirmed by the formation of brown-coloured precipitate.

Figure 2B presents the influence of pH value of polymerization solution on the formation of PANI/GOx composite nanoparticles. It was determined that, by the increase of pH value from 1.0 to 10, the optical absorbance of formed colloidal PANI/GOx nanoparticle solution at *λ*_max_ = 440 nm has decreased 5.52 times (from 0.690 to 0.125 a.u.). The highest value of optical absorbance (0.690 a.u. at *λ*_max_ = 440 nm) was registered in the solution of pH 1.0 after 48 h lasting polymerization of aniline. However, the pH 1.0 is not suitable for PANI/GOx formation due to the inactivation of GOx and due to the non-enzymatic polymerization at this extreme pH value. In the research that was performed by another authors, it was presented that, by the increase of pH value, the polaron optical absorbance at *λ*_max_ = 420 nm and *λ*_max_ = 823 nm gradually decreases and a strong optical absorbance is observed due to the transition of the quinoid rings, which emerges at *λ* = 560–600 nm [22]. The most suitable solution for the formation of emeraldine form of PANI/GOx composite nanoparticles was SA buffer solution, pH 6.0 (absorbance value of 0.329 a.u. at *λ*_max_ = 440 nm was observed).

The relationship between the optical absorbance of colloidal solution of Ppy/GOx composite nanoparticles and the pH values in the range from 2.0 to 12 of polymerization solution is presented in Figure 3. The optical absorbance peak at *λ*_max_ = 460 nm has been observed during the formation of Ppy/GOx composite nanoparticles, which is in an agreement with the results that were obtained during the chemical polymerization of pyrrole [7]. Optical absorbance at *λ*_max_ = 460 nm is attributed to a transition from valence bond to the polaron or bipolaron state [7]. The peak of optical absorbance with *λ*_max_ = 460 nm was observed at the early stage of pyrrole oxidation, and it was identified that it belongs to the intermediate that formed during the oxidative polymerization of pyrrole. Some authors mentioned the optical absorbance peak at *λ*_max_ = 360 nm, which is related to *π*-*π** transition [7]. During enzymatic polymerization, the formed Ppy/GOx composite nanoparticles were of intensive black colour when polymerization was performed at pH 2.0 and pH 3.0; and, intensive dark-grey—in pH range from 5.0 to 12. Black colour is the most characteristic for the delocalised *π*-electron system of doped Ppy [25].

It was determined that, by the increase of pH value of polymerization bulk solution from 3.0 to 6.0, the absorbance at *λ*_max_ = 460 nm has increased by 1.60 times (Figure 3B). Additionally, the decrease of the absorbance was followed by the increase of pH value from 8.0 to 12, and it is in agreement with investigations of other researchers [18]. The absorbance at *λ*_max_ = 460 nm has decreased by 9.17 times when polymerization was performed in pHs ranging from 8.0 to 12 (Figure 3B). Alkaline solutions were considered to be less suitable for Ppy formation. According to investigations [18], the neutral radical that formed in strong bases is not able to form the dimer that consists of two pyrrole molecules. The highest value of the absorbance (0.207 a.u.) was registered at pH 6.0 after 48 h of enzymatic polymerization in all cases.

The main charge carriers in PTh are polarons and bipolarons [15]. Figure 4 presents the optical absorbance spectra of formed PTh/GOx composite nanoparticles in the pH interval from 1.0 to 12. The UV/Vis absorbance spectra of colloidal solutions of PTh/GOx are characterized by optical absorbance peak at *λ*_max_ = 450 nm. Optical absorbance peak (*λ*_max_ = 450 nm) is attributed to the *π*-*π** transition for large conjugated structure [12,34]. Some authors mentioned optical absorbance peak at *λ*_max_ = 890 nm, which is related to the bipolaron state [15]. The appearance of optical absorbance peak at *λ*_max_ = 450 nm was in agreement with the investigations related to chemical-oxidative formation of: (i) PTh-based 60–100 nm nanoparticles in water solution containing copper (II) ions and H_2_O_2_ as oxidizer [12] and (ii) PTh-based 25–45 nm nanoparticles in the presence of FeCl_3_ as an oxidizer [15].

As presented in Figure 4B, the absorbance at *λ*_max_ = 450 nm was dramatically changed (4.96 times), from 0.124 to 0.025 a.u. by the increase of pH value from 1.0 to 12. The absorbance of 0.023 a.u. at *λ*_max_ = 450 nm was registered at pH 6.0 after 48 h of enzymatic polymerization.

### 3.2. The Comparison of Optical Absorbance of PANI/GOx, Ppy/GOx and PTh/GOx Composite Nanoparticles Colloidal Solution and the Choice of Optimal Enzyme/Substrate Ratio

The optical absorbance of PANI/GOx, Ppy/GOx, and PTh/GOx composite nanoparticles colloidal solutions at *λ*_max_ = 440 nm, *λ*_max_ = 460nm, and *λ*_max_ = 450 nm, respectively, was assessed in SA buffer, pH 6.0, after 48 h lasting polymerization and was 0.329, 0.207, and 0.023 a.u., respectively (Figure 5A). It was noticed that the optical absorbance of PTh/GOx composite nanoparticles after 48 h lasting enzymatic polymerization was 14.3 and 9.00 times lower than that for the PANI/GOx and Ppy/GOx composite nanoparticles. As it is presented in Figure 5A, the absorbance of formed PANI/GOx colloidal solution after 48 h of polymerization was 1.59 times higher than that of the Ppy/GOx colloidal solution. The optical control of PTh/GOx composite nanoparticles formation was more complicated in comparison to that of PANI/GOx or Ppy/GOx composite nanoparticles formation due to the slower polymerization rate. It should be noted that it is quite difficult to prepare the solution of well-dispersed polythiophene particles, because they tend to aggregate and precipitate [12,34]. Therefore, further investigations of a polymerization’s duration were focused on the enzymatic formation of PANI/GOx and Ppy/GOx composite nanoparticles.

As it has been mentioned by other authors, the Ppy formation can be performed in the solution without any chemical oxidant or enzyme [18], and this process is named autopolymerization. An experiment was performed in solutions containing only 0.05 mol L^−1^ of glucose and 0.5 mol L^−1^ of aniline, pyrrole, or thiophene at pH from 1.0 to 12 without GOx, to evaluate the influence of the autopolymerization on the formation of PANI, Ppy and PTh. Autopolymerization was not observed in the pH range from 1.0 to 12 for thiophene. The slow autopolymerization has only been observed at pH 1.0 for aniline and at pH below 2.0 for pyrrole and it reached 0.030 and 0.035 a.u. after 48 h lasting polymerization, respectively. This autopolymerization phenomenon was not observed at a higher pH value. The polymerization rate of aniline was 1.20 times slower in the comparison of pyrrole at pH 2.0 at pH 1.0 in the absence of GOx. The polymerization rate of pyrrole at pH 2.0 in the absence and in the presence of GOx was the same. It means that GOx was inactivated at this low pH. We have performed the polymerization at pH values higher than 3.0 in order to avoid the autopolymerization. The results of enzymatic polymerization were in an agreement with the results that were obtained for PANI [21] and Ppy [24]. Significant differences of the absorbance were detected in the absence and in the presence of GOx demonstrate a significant impact of the enzyme to polymerization reaction rate. The most optimal formation rate of PANI/GOx and Ppy/GOx composite nanoparticles was observed in 0.05 mol L^−1^ SA buffer, pH 6.0, where GOx exhibits the maximal catalysed reaction rate and the highest stability, according to our measurements [24,39].

The type of oxidizing agent has significant influence on the oxidative polymerization reaction [18]. It was determined that the polymerization is faster at higher (0.50 mol·L^−1^) monomer’s concentration [22], but further increasing of monomer’s concentration negatively influenced the formation of polymeric nanoparticles. The rate of polymerization reaction and the properties of the formed polymer not only depend on the preparation technique, selected solvent, polymerization duration, temperature, kind, and concentration of monomer, but also on the enzyme and substrate ratio [18,21].

The effect of enzyme/substrate ratio on the rate of PANI/GOx, Ppy/GOx, and PTh/GOx composite nanoparticles formation was investigated in the next set of spectrophotometric measurements. The investigations were performed during 48 h in the polymerization solution, pH 6.0, containing 0.50 mol L^−1^ of aniline, pyrrole, or thiophene, 0.75 mg mL^−1^ of GOx and glucose concentration varying from 0.015 to 0.05 mol L^−1^. Figure 5B presents the influence of the ratio of GOx/Glucose on the registered absorbance.

The registered absorbance significantly increased by increasing glucose concentration from 0.015 mol L^−1^ in the polymerization solution up to 0.05 mol L^−1^. The absorbance of PANI/GOx, Ppy/GOx, and PTh/GOx composite nanoparticles, which were prepared using 0.75 mg mL^−1^ of GOx and 0.05 mol L^−1^ of glucose, increased by 3.39, 3.79, and 10.1 times in a comparison to the absorbance that was obtained for polymeric nanoparticles with embedded GOx using 0.015 mol L^−1^ of glucose. By the increase of glucose concentration up to 0.015 mol L^−1^, the absorbance of PANI/GOx, Ppy/GOx, and PTh/GOx decreased 1.25, 1.31, and 3.53 times, if compared with the results that were obtained for the absorbance with 0.75 mg mL^−1^ of GOx and 0.05 mol L^−1^ of glucose. Thus, the ratio of GOx/Glucose 0.75 mg mL^−1^/0.05 mol L^−1^ was selected as the most optimal for polymeric nanoparticle composites formation.

### 3.3. The Evaluation of PANI/GOx, Ppy/GOx and PTh/GOx Composite Nanoparticles Formation by DLS and the Morphology of PANI/GOx, Ppy/GOx Nanocomposites

Depending on pH, the negatively or positively charged enzymes can be entrapped within the CPs during oxidation/reduction reactions [20]. Enzymatic polymerization strongly depends on the solution’s pH. For instance, the conducting form of linear PANI is synthesized at pH 4.0–4.5, wherein, at pH 6.0 or higher, more branched and insulating form of PANI is formed [11,22].

The diameter of 48 h enzymatically synthesized PANI-, Ppy-, and PTh-based nanocomposites with embedded GOx in solution of pH 2.0–8.0 was evaluated by the DLS technique. The results of PANI/GOx, Ppy/GOx, and PTh/GOx composite nanoparticles diameter and their distribution are presented in Table 1 and Figure 6, respectively. The diameter of PANI/GOx, Ppy/GOx, and PTh/GOx composites nanoparticles depended on 5h3 pH of polymerization solution and it decreased by the increase of pH. It was determined that, in SA buffer, pH 6.0, PANI/GOx, Ppy/GOx, and PTh/GOx composite nanoparticles of 54, 86, and 98 nm diameter were formed, respectively. The DLS results (Table 1) represent that formed PANI/GOx composite nanoparticles were 1.59 and 1.81 times smaller if compared with Ppy/GOx and PTh/GOx. Due to the relatively large (535 nm) diameters of PTh/GOx, which were formed at pH 8.0, we decided that such relatively large particle colloidal solution will be not very stable and mostly focused on the investigation of PANI/GOx and Ppy/GOx composite nanoparticles, which are smaller; therefore, it was expected that colloidal solution of these nanoparticles will be more stable.

Some of the researchers evaluated the diameter of synthesized polymer nanocomposites by FE-SEM. During the enzymatic polymerization of aniline using chitosan and poly(*N*-isopropylacrylamide) as steric stabilizers 50 nm PANI nanoparticles were synthesized [11]. In our study, PANI/GOx and Ppy/GOx composite nanoparticles formed during 24 h lasting enzymatic polymerization were investigated by FE-SEM (Figure 7).

Spherical PANI/GOx and Ppy/GOx composite nanoparticles were formed during enzymatic polymerization. It is seen that the PANI/GOx (Figure 7A) nanoparticles were randomly distributed on the surface of graphite rod used for investigations. The Ppy/GOx nanoparticles were densely stuck together (Figure 7B). The diameter of nanoparticles depends on the kind of polymerized monomer. It was determined that the diameter of PANI/GOx and Ppy/GOx composites was in the range of 38–50 and 50–100 nm, respectively.

### 3.4. The Influence of Polymerization Duration on PANI/GOx and Ppy/GOx Composite Nanoparticles Formation

One of most important parameters of polymer‘s formation is the duration of a polymerization. The influence of polymerization duration (from 5 to 288 h) on the formation of PANI/GOx and Ppy/GOx composite nanoparticles was investigated. The polymerization was performed in 0.05 mol L^−1^ SA buffer, pH 6.0, with 0.05 mol L^−1^ of glucose, 0.50 mol L^−1^ of aniline or pyrrole, and 0.75 mg mL^−1^ of GOx. Visible aggregation of PANI/GOx and Ppy/GOx was observed after 48 h lasting enzymatic polymerization. Monomers are oxidized during the enzymatic polymerization and the amount of polymers is increasing by the prolongation of synthesis’s duration. It should be noted that the enzyme tends to loss catalytic activity due to long lasting polymerization [18]. The amount and the intensity of absorbance for the obtained PANI/GOx and Ppy/GOx composite nanoparticles at *λ*_max_ = 440 nm and *λ*_max_ = 460 nm increased by the prolongation of polymerization duration from 5 to 168 h (Figure 8).

Such an effect indicated the formation of PANI/GOx and Ppy/GOx composite nanoparticles and it is in an agreement with some previous researches [18,24]. The maximal concentration of PANI/GOx and Ppy/GOx composite nanoparticle formation was achieved after 168 h. Optical absorbance after 168 h at *λ*_max_ = 440 nm and *λ*_max_ = 460 nm has increased by 74.7 and 11.1 times, respectively, in a comparison with the absorbance that was registered after 5 h (Figure 8). After 168 h lasting formation of Ppy/GOx composite nanoparticles the optical absorbance at *λ*_max_ = 460 nm was 2.48 times lower than that of PANI/GOx at *λ*_max_ = 440 nm after the same duration of polymerization. It was determined that the absorbance of PANI/GOx and Ppy/GOx composite nanoparticles has decreased during long-lasting polymerization (more than 168 h). The limited solubility of synthesized polymers and the precipitation of particles or clusters of PANI/GOx and Ppy/GOx from the polymerization solution could explain this phenomenon [18]. When the formation of PANI/GOx and Ppy/GOx composites lasted longer than 168 h, the significant part of formed nanoparticles has precipitated. Though, optical absorbance at *λ*_max_ = 440 nm and *λ*_max_ = 460 nm of PANI/GOx and Ppy/GOx after 48 h lasting polymerization was 2.52 and 1.94 times lower than that registered after 168 h lasting polymerization. 48–96 h lasting polymerization seems to be more efficient for PANI/GOx and Ppy/GOx composite nanoparticles formation, because smaller PANI/GOx and Ppy/GOx will be formed.

The initial aniline and pyrrole conversion into polymer reaction rate (*V*) was evaluated by formula *V* = (tgα), where tgα was calculated from Figure 8 by the evaluation of estimated/approximated absorbance of PANI/GOx and Ppy/GOx solutions (1.27 a.u. for PANI/GOx and 0.446 a.u. for Ppy/GOx) that formed when 100% of monomers, which were present in the solution, will be converted into polymer. The initial aniline and pyrrole polymerization rate was determined as 0.0037 and 0.0106 mol L^−1^ h^−1^, respectively. It is seen that the initial polymerization rate of aniline is 2.86 times slower than that of pyrrole.

The diameter of PANI/GOx and Ppy/GOx composite nanoparticles and their aggregates was investigated by DLS and Figure 9 and Table 2 present the results. The diameter of polymeric nanoparticles increased by the increase of polymerization duration from 16 to 120 h. 41 nm PANI/GOx and 65 nm Ppy/GOx were formed after 16 h of enzymatic polymerization. The diameter of PANI/GOx and Ppy/GOx increased up to 167 and 122 nm, respectively, after 120 h of the polymerization.

The aggregates of PANI/GOx and Ppy/GOx composite nanoparticles were formed from the 120th hour of enzymatic polymer’s formation. During 168–288 h lasting polymerization, a lot of agglomerates that are characterized by large diameter were formed, which was in an agreement with our previous researches, where 142 nm PANI/GOx composite nanoparticles were formed after 168 h lasting polymerization [21]. For the enzymatic synthesis of smaller polymeric nanoparticles, 48 h lasting polymerization is the most suitable, because, in such way, 72 nm PANI/GOx (Figure 9B) and 43 nm Ppy/GOx composite nanoparticles were formed. Other authors have reported that, during enzymatic polymerization of aniline using chitosan and poly(*N*-isopropylacrylamide) as steric stabilizers and horseradish peroxidase as enzyme, 50 nm PANI nanoparticles were synthesized [11]. However, the PANI matrix doped by small anions is characterized by poor stability. Therefore, it is recommended to dope PANI by larger anions to solve this problem [28].

The electrical charge of particles is usually expressed by zeta-potential. The zeta-potential was determined in SA buffer, pH 6.0, for polymeric nanoparticles formed by 48 h lasting polymerization in our investigations. The zeta-potentials of PANI/GOx, Ppy/GOx, and PTh/GOx nanoparticles were −6.2, −9.4, and −8.6 mV, respectively.

## 4. Conclusions

PANI/GOx, Ppy/GOx, and PTh/GOx composite nanoparticles were synthesized by the enzymatic polymerization of aniline, pyrrole, and thiophene, respectively. The PANI/GOx, Ppy/GOx, and PTh/GOx nanoparticles were characterized by optical absorbance maximums at *λ*_max_ = 440 nm, *λ*_max_ = 460 nm, and *λ*_max_ = 450 nm, respectively. The formation of PANI/GOx in emeraldine base form was observed. Both factors (i) the pH of polymerization bulk solution and (ii) the duration of polymerization have significant influence on the formation of polymeric nanoparticles. The highest rate and the smallest diameter of formed PANI/GOx, Ppy/GOx, and PTh/GOx were observed in solution of sodium acetate buffer, pH 6.0. The optimal ratio of glucose oxidase/glucose for the formation of PANI/GOx, Ppy/GOx, and PTh/GOx composite nanoparticles was determined as 0.75 mg mL^−1^/0.05 mol L^−1^. Spherical particles of PANI/GOx and Ppy/GOx were formed during enzymatic polymerization. The most optimal duration for the enzymatic synthesis of PANI/GOx and Ppy/GOx composite nanoparticles was in the range of 48–96 h. The diameter of polymeric nanoparticles, which was determined by DLS, was dependent on the duration of polymerization: 41–167 nm PANI/GOx and 65–122 nm Ppy/GOx particles were formed by polymerization lasting from 16 to 120 h, respectively. The formed PANI/GOx and Ppy/GOx composite nanoparticles were pure from surfactants and other interfering species, which increases the applicability of synthesized polymeric composite nanoparticles for possible biomedical applications.

Here, the evaluated PANI/GOx and Ppy/GOx composite nanoparticles formation conditions can be applied in further researches. Some advantages of the application of gold nanoparticles in the formation of conducting polymers-based composite materials have been demonstrated [37,39]. Therefore, we are predicting that the extension of PANI/GOx and Ppy/GOx applicability in biosensorics can be achieved by embedment of gold nanoparticles within PANI/GOx and Ppy/GOx composite structure.

## Figures and Tables

**Figure 1 nanomaterials-09-00806-f001:**
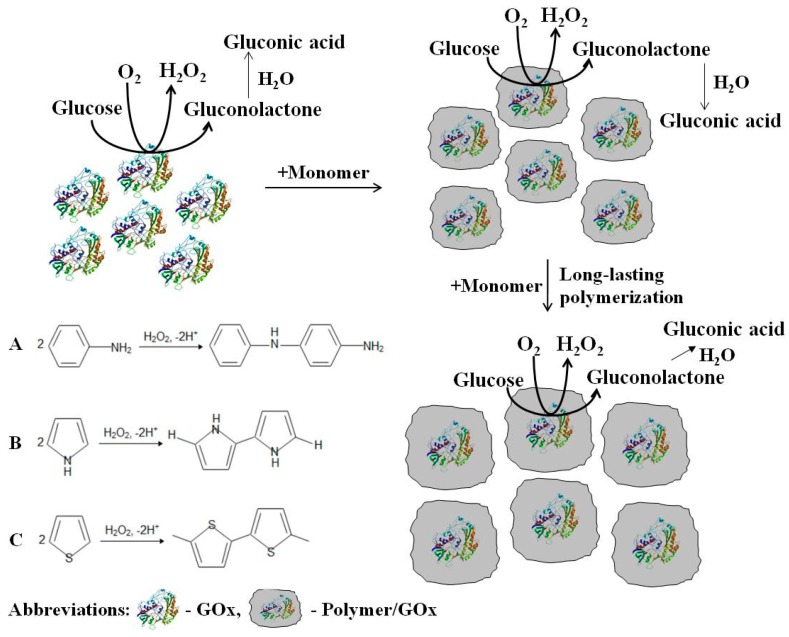
The formation of composite conducting polymer nanoparticles with embedded glucose oxidase (GOx) by enzymatic polymerization and the schematic presentation of polyaniline (PANI) (**A**), polypyrrole (Ppy) (**B**), and polythiothene (PTh) (**C**) polymerization mechanisms.

**Figure 2 nanomaterials-09-00806-f002:**
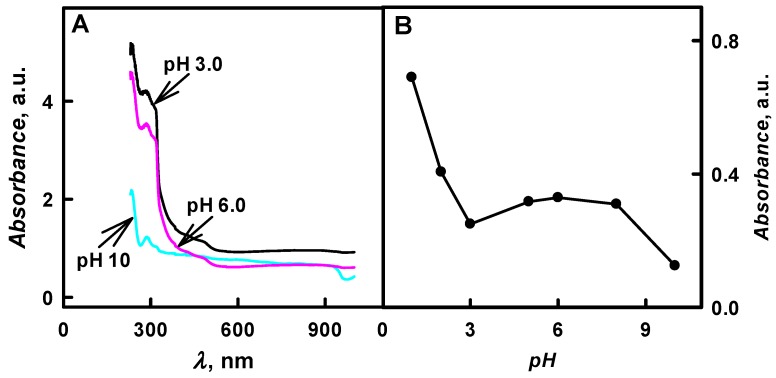
(**A**) Spectra of colloidal solutions of PANI/GOx composite nanoparticles synthesized at various solution pH values and (**B**) the influence of polymerization bulk solution pH on the absorbance at *λ*_max_ = 440 nm. (The composition of polymerization solution: 0.05 mol L^−1^ of glucose, 0.50 mol L^−1^ of aniline and 0.75 mg mL^−1^ of glucose oxidase at pH 1.0–10; 48 h lasting polymerization was performed. Optical absorbance was registered in 0.05 mol L^−1^ SA buffer, pH 6.0).

**Figure 3 nanomaterials-09-00806-f003:**
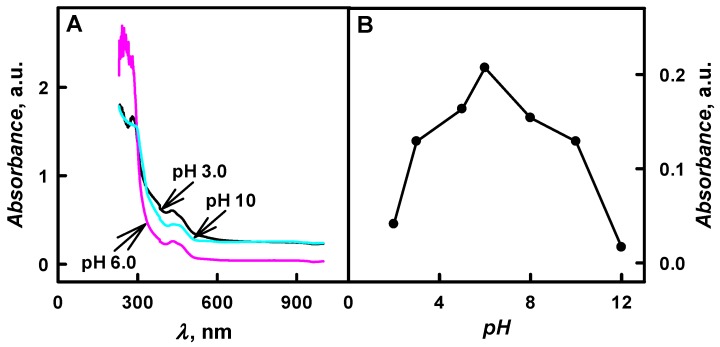
(**A**) The spectra of formed Ppy/GOx composite nanoparticles colloidal solution at various solution pH values and (**B**) dependence of optical absorbance at *λ*_max_ = 460 nm on the solution pH. (The composition of polymerization solution: 0.05 mol L^−1^ of glucose, 0.50 mol L^−1^ of pyrrole, and 0.75 mg mL^−1^ of glucose oxidase at pH 2.0–12; 48 h lasting polymerization was performed. Optical absorbance was registered in 0.05 mol L^−1^ SA buffer, pH 6.0).

**Figure 4 nanomaterials-09-00806-f004:**
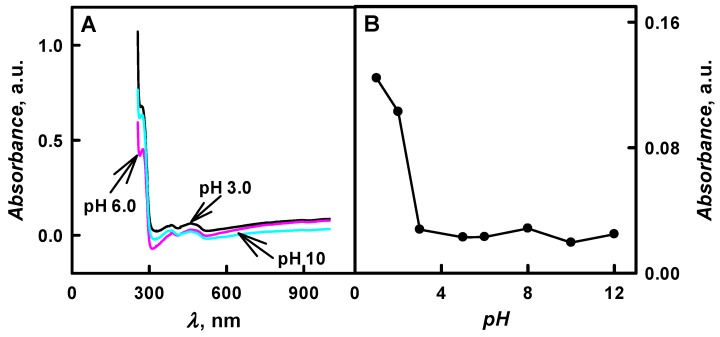
(**A**) Spectra of PTh/GOx composite nanoparticles colloidal solutions formed at various solution pH values and (**B**) dependence of optical absorbance at *λ*_max_ = 450 nm on the solution pH. (The composition of polymerization solution: 0.05 mol L^−1^ of glucose, 0.50 mol L^−1^ of thiophene, and 0.75 mg mL^−1^ of glucose oxidase at pH 1.0–12; 48 h lasting polymerization was performed. Optical absorbance was registered in 0.05 mol L^−1^ SA buffer, pH 6.0).

**Figure 5 nanomaterials-09-00806-f005:**
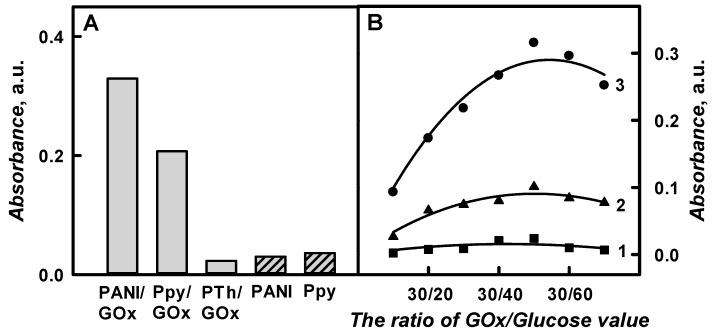
The diagram of optical absorbance of PANI-, Ppy-, and PTh-based composite nanoparticles in the presence and absence of GOx (**A**) and the influence of GOx/Glucose ratio on the absorbance for PTh/GOx (1 curve), Ppy/GOx (2 curve), or PANI/GOx (3 curve) composite nanoparticles colloidal solutions at pH 6.0 (**B**). ((**A**) The composition of polymerization solution, pH 6.0: 0.05 mol L^−1^ of glucose, 0.50 mol L^−1^ of aniline, pyrrole, or thiophene and 0.75 mg mL^−1^ of glucose oxidase (grey columns); the composition of solution used for autopolymerization: 0.05 mol L^−1^ of glucose, 0.50 mol L^−1^ of aniline or pyrrole at pH 1.0 or pH 2.0, respectively (dash columns); (**B**) The composition of polymerization solution, pH 6.0: 0.50 mol L^−1^ of aniline, pyrrole or thiophene, 0.75 mg mL^−1^ of glucose oxidase and 0.015–0.05 mol L^−1^ of glucose. Polymerization lasted for 48 h. Optical absorbance was registered for PANI/GOx and PANI, Ppy/GOx, and Ppy, PTh/GOx at *λ*_max_ = 440 nm, *λ*_max_ = 460 nm, and *λ*_max_ = 450 nm in 0.05 mol L^−1^ SA buffer, pH 6.0).

**Figure 6 nanomaterials-09-00806-f006:**
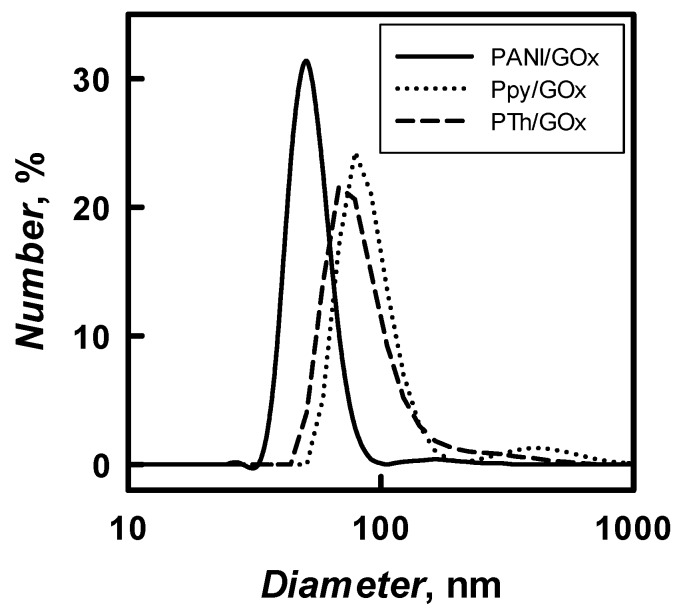
The distribution of PANI/GOx, Ppy/GOx, and PTh/GOx composite nanoparticles diameter. The particles were formed by 48 h lasting enzymatic polymerization at pH 6.0.

**Figure 7 nanomaterials-09-00806-f007:**
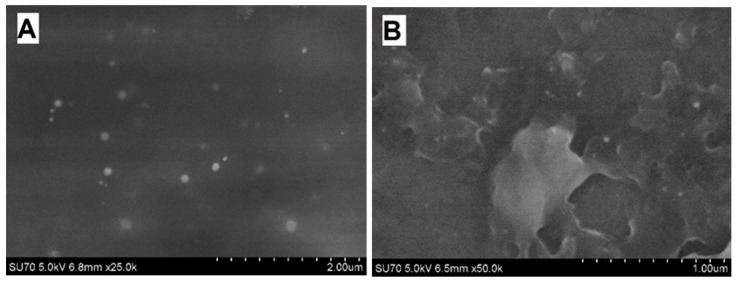
FE-SEM images of PANI/GOx (**A**) and Ppy/GOx (**B**) composite nanoparticles after 24 h lasting enzymatic polymerization. (The polymerization solution: 0.05 mol L^−1^ SA buffer, pH 6.0, with 0.05 mol L^−1^ of glucose, 0.75 mg mL^−1^ of GOx, and 0.50 mol L^−1^ of aniline or pyrrole).

**Figure 8 nanomaterials-09-00806-f008:**
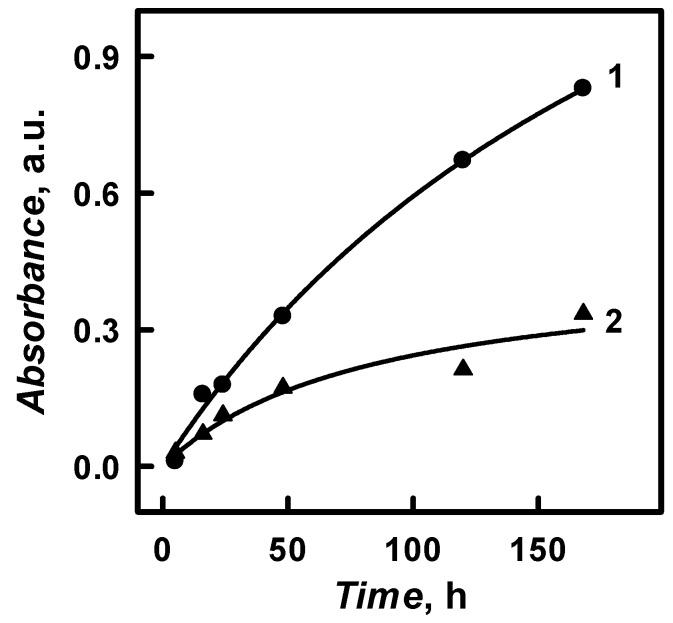
The influence of polymerization duration on PANI/GOx and Ppy/GOx composite nanoparticles optical absorbance. (The composition of polymerization solution: 0.05 mol L^−1^ SA buffer, pH 6.0, with 0.05 mol L^−1^ of glucose, 0.50 mol L^−1^ of aniline, or pyrrole and 0.75 mg mL^−1^ of glucose oxidase. Optical absorbance was registered in 0.05 mol L^−1^ SA buffer, pH 6.0, at *λ*_max_ = 440 nm, and *λ*_max_ = 460 nm for PANI/GOx and Ppy/GOx, respectively).

**Figure 9 nanomaterials-09-00806-f009:**
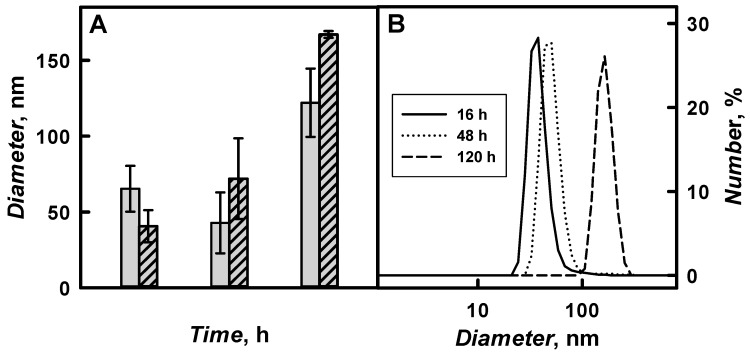
The influence of enzymatic polymerization duration on the diameter of formed PANI/GOx (dash column) and Ppy/GOx (grey column) composite nanoparticles (**A**) and the distribution of PANI/GOx nanoparticles diameter after 16, 48, and 120 h polymerization (**B**). (The composition of polymerization solution: 0.05 mol L^−1^ SA buffer, pH 6.0, with 0.05 mol L^−1^ of glucose, 0.50 mol L^−1^ of aniline or pyrrole, and 0.75 mg mL^−1^ of glucose oxidase. DLS signal was registered in 0.05 mol L^−1^ SA buffer, pH 6.0).

**Table 1 nanomaterials-09-00806-t001:** The dependence of PANI/GOx, Ppy/GOx, and PTh/GOx composite nanoparticles diameter on the pH of solutions. The composition of polymerization solution: 0.05 mol L^−1^ of glucose, 0.75 mg mL^−1^ of GOx and 0.50 mol L^−1^ of aniline, pyrrole, or thiophene. The particles were formed by 48 h lasting enzymatic polymerization. The diameter of particles was determined by dynamic light scattering (DLS) in 0.05 mol L^−1^ SA buffer, pH 6.0.

*pH*	*Diameter*, nm		
	PANI/GOx	Ppy/GOx	PTh/GOx
2.0	180 ± 94	180 ± 46	169 ± 62
5.0	56 ± 22	155 ± 61	63 ± 9.4
6.0	54 ± 6.9	86 ± 18	98 ± 28
8.0	106 ± 41	308 ± 49	535 ± 106

**Table 2 nanomaterials-09-00806-t002:** The characteristics of PANI/GOx and Ppy/GOx composite nanoparticles depending of the polymerization duration determined by DLS. (Conditions like in Figure 9, SD—standard deviation.).

*Time*, h	PANI/GOx		Ppy/GOx	
	*Diameter*, nm	SD	*Diameter*, nm	SD
16	41	±11	65	±15
48	72	±27	43	±20
120	167	±28	122	±23

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
