# Peer review of "Enzymatic Formation of Polyaniline, Polypyrrole, and Polythiophene Nanoparticles with Embedded Glucose Oxidase"

_nanomaterials, 2019, doi:10.3390/nano9050806_

Reviewer 1 Report

It seems that the authors have improved their manuscript as asked by the reviewer. However, there is no "Answers to the reviewers' comments" file. Moreover, the added paragraph on page 2 needs to be carefully read to be improved. Indeed, this part is quite difficult to understand.

However, I do recommand the publication of this manuscript in "Nanomaterials" after minor changes.

Author Response

Response to reviewer #1:

We would like to thank the reviewer for very professional review of our manuscript, valuable comments and recommendations. Thank you for pointing out our mistakes and giving suggestions which further on improve clarity of this paper. We did our best in order to improve the manuscript according to comments and recommendations. All the most important changes are highlighted in the revised manuscript. Corrections and changes are highlighted in the manuscript (in red).

Please   find below short explanations and answers to your questions during 1st  round of evaluation, additional corrections   after second round are indicated in red:

Reviewer #1 wrote: Abstract: what about the results obtained for the PTh-based nanoparticles? The authors have to add few words on these formulations: they mentioned them in the title and the beginning of the abstract.

Response to Reviewer #1: Abstract: the information about PTh/GOx nanoparticles formation was added:

18 line the phrase “the duration” was deleted and 18-20 lines were rewritten toThe influence of used monomer, the pH of solution and the ratio of enzyme/substrate on the formation of PANI/GOx, Ppy/GOx and PTh/GOx composite nanoparticles was investigated spectrophotometrically.

20-22 lines were rewritten toThe highest formation rate of PANI-, Ppy- and PTh-based nanoparticles with embedded GOx was observed in the solution of sodium acetate (SA) buffer, pH 6.0.

22-23 lines were rewritten toThe increase of optical absorbance at lmax = 440 nm, lmax = 460 nm and lmax = 450 nm was exploited for the monitoring of PANI/GOx, Ppy/GOx and PTh/GOx formation, respectively.

26-27 lines, the sentenceThe influence of the enzymatic polymerization duration on the formation of PANI/GOx and Ppy/GOx composite nanoparticles was investigated spectrophotometrically.was added.

29-31 lines were rewritten toUsing dynamic light scattering technique (DLS) it was determined that the diameter of formed PANI/GOx and Ppy/GOx composite nanoparticles depends on the duration of polymerization and it was in the range of 40.6–167 nm and 65.3–122 nm, when polymerization was from 16 to 120 h.

Reviewer #1 wrote:  Introduction, line 3: the word ”, respectively” has to be added at the end of the sentence “The electrical conductivity … S cm-1” just before the references.

Response to Reviewer #1: Introduction, 39 line (was 34 line): the word , respectivelywas added at the end of the sentence “The electrical conductivity … S cm-1” before the references.

Reviewer #1 wrote:  Introduction, line 36: the word “stabile” has to be changed to “stable”.

Response to Reviewer #1: Introduction, 41 line (was 36 line): the word “stabile” was changed tostable”.

Reviewer #1 wrote:  Introduction, line 43 : “A novel strategies” has to be changed to “Novel strategies”

Response to Reviewer #1: Introduction, 49 line (was 43 line): “A novel strategies” was changed to Novel strategies”.

Reviewer #1 wrote:  Introduction, lines 55 to 61: these sentences are not clear and have to be reformulated. Moreover, what happen when the pH is in the range of 2 and 4.6? Do the authors think it can be possible to apply such low pH for an enzymatic reaction? Don’t they think that the enzyme will be inactivated at pH 1?

Response to Reviewer #1: Introduction, 60-70 lines (were 55-61 lines): these sentences were reformulated more clearly and process in the range pHs from 2.5 to 4.6 was described:

Conducting (4-10 S cm-1) PANI is produced during chemical polymerization in the solution of high acidity (pH < 2.5) and the morphology of formed composite depends on the pH value [5,8,19]. In the pH range from 2.5 to 4.6 considerable part of aniline is protonated, but imino groups in polymeric chains still remain unprotonated [6]. At these pHs oxidative processes are attenuated and the rate of proton release is decreased [6]. Under the conditions of low acidity (pH 4.6) PANI is present in the form of neutral molecules and is able to oxidize more easier than anilinium cation, which is dominant at higher acidity [5,28]. When the polymerization reaction is occurring at pH > 4.6, then oxidized terminal imino groups of PANI replaces hydrogen atom in benzene ring of the neutral aniline molecule [6]. At pH value higher than 4.6 the PANI is formed not in regular structure, but in the form of oligomers with different structures, based on orto- and para-coupled units with cyclic phenazine fragments [6,19].

Of course, the pHs 1 and 2 will be not use for enzymatic polymerization, because at low pH value enzyme will be inactivated. For enzymatic formation of PANI/GOx composite nanoparticles neutral polymerization solution is required due to best enzymatic activity and stability of GOx [21].

Reviewer #1 wrote:  Introduction, lines 89 to 90: Same remark as the 1st one, what about PTh/Gox materials? Don’t they form nanoparticles? In the experimental part, the authors described the formation of Pth/Gox based nanoparticles?

Response to Reviewer #1: PTh/GOx nanoparticles was formed enzymatically. The investigation of PTh/GOx composite was performed by spectrometric and DLS analysis. The influence of enzymatic polymerization time wasn’t investigated because the main purpose of previously investigations was the choice of more suitable polymeric nanoparticles. In our case it was PANI/GOx and Ppy/GOx nanoparticles.

Introduction: the information about PTh/GOx nanoparticles formation and investigation was added:

107-109 lines (were 88-89 lines) were rewritten to “The effect of pH, the ratio of enzyme/substrate on the formation of these composite nanoparticles and the influence polymerization duration on the formation of PANI/GOx, Ppy/GOx were investigated by UV/Vis spectroscopy.”

109-110 lines (were 89-90 lines): the information about PTh/GOx nanoparticles formation was added and the sentence was rewrittenThe diameter of PANI/GOx, Ppy/GOx and PTh/GOx nanoparticles was evaluated using dynamic light scattering (DLS) technique.

Materials and methods: 159-172 lines were supplemented by the information about DLS and zeta-potential for PANI/GOx, Ppy/GOx and PTh/GOx:

2.5. The evaluation of PANI/GOx, Ppy/GOx and PTh/GOx composite nanoparticles formation by DLS

The influence of solution’s pH on the diameter and zeta-potential of PANI/GOx, Ppy/GOx and PTh/GOx composite nanoparticles was studied in pH range from 2.0 to 8.0 at +20 ± 2 °C. In this experiment 0.01 or 0.001 mol L1 HCl, 0.05 mol L1 SA buffer, pH 6.0, and 1 × 106 mol L1 NaOH were used to prepare solutions in pH range between 2.0 and 8.0. Polymerization solutions containing 0.05 mol L1 of glucose, 0.50 mol L1 of aniline, pyrrole or thiophene and 0.75 mg mL1 of GOx were monitored during 15-120 h lasting polymerization.

Polymeric nanoparticles after enzymatic synthesis were separated, washed two times with 0.05 mol L−1 SA buffer, pH 6.0, and collected by the centrifugation (6 min, 16.1 × 103 g) as it is described in chapter 2.2. Diameter and zeta-potential of PANI/GOx, Ppy/GOx and PTh/GOx composite nanoparticles were evaluated by DLS and microelectrophoresis, using the Zetasizer Nano ZS from Malvern (Herrenberg, Germany) equipped with a 633 nm He-Ne laser. The obtained data was analysed with Dispersion Technology Software version 6.01 from Malvern. DLS and microelectrophoresis investigations were evaluated and visualized by SigmaPlot software 12.5.

Results and discussion: 370-371 lines were supplemented 3.3. The evaluation of PANI/GOx, Ppy/GOx and PTh/GOx composite nanoparticles formation by DLS and the morphology of PANI/GOx, Ppy/GOx nanocomposites.”

Results and discussion, 372-397 lines: paragraphs of investigations by DLS were supplemented:

Depending on pH the negatively or positively charged enzymes can be entrapped within the CPs during oxidation/reduction reactions [20]. Enzymatic polymerization is strongly depending on solutions pH. For instance, the conducting form of linear PANI is synthesized at pH 4.0-4.5 , wherein at pH 6.0 or higher – more branched and insulating form of PANI is formed [11,22].

The diameter of 48 h enzymatically synthesized PANI-, Ppy- and PTh-based nanocomposites with embedded GOx in solution of pH 2.0-8.0 was evaluated by DLS technique. Results of PANI/GOx, Ppy/GOx and PTh/GOx composite nanoparticles diameter and their distribution are presented in Table 1 and figure 6, respectively. The diameter of PANI/GOx, Ppy/GOx and PTh/GOx composites nanoparticles depended on pH of polymerization solution and it decreased by the increase of pH. It was determined that in SA buffer, pH 6.0, PANI/GOx, Ppy/GOx and PTh/GOx composite nanoparticles of 53.9, 85.8 and 98.2 nm diameter were formed, respectively. DLS results (Table 1) represent that formed PANI/GOx composite nanoparticles were by 1.59 and 1.82 times smaller if compared with Ppy/GOx and PTh/GOx. Due to relatively large (535 nm) diameter of PTh/GOx, which were formed at pH 8.0, we decided that such relatively large particle colloidal solution will be not very stable and focussed mostly on the investigation of PANI/GOx and Ppy/GOx composite nanoparticles, which are smaller, therefore it was expected that colloidal solution of these nanoparticles will be more stable.

Table 1. The dependence of PANI/GOx, Ppy/GOx and PTh/GOx composite nanoparticles diameter on the pH of solutions. The composition of polymerization solution: 0.05 mol L-1 of glucose, 0.75 mg mL-1 of GOx and 0.50 mol L-1 of aniline, pyrrole or thiophene. Particles were formed by 48 h lasting enzymatic polymerization. The diameter of particles was determined by DLS in 0.05 mol L-1 SA buffer, pH 6.0. 

pH

Diameter, nm

PANI/GOx

Ppy/GOx

PTh/GOx

2.0

180 ±94.3

180 ±45.5

   169 ±61.6

5.0

56.1 ±21.8

155 ±61.0

62.5 ±9.36

6.0

53.9 ±6.86

85.8 ±17.7

98.2 ±28.0

8.0

106 ±41.3

308 ±48.5

535 ±106

Figure 6. The distribution of PANI/GOx, Ppy/GOx and PTh/GOx composite nanoparticles diameter. Particles were formed by 48 h lasting enzymatic polymerization at pH 6.0.

Results and discussion, 480-483 lines: paragraph of zeta-potential investigations was supplemented:

The electrical charge of particles is usually expressed by zeta-potential. In our investigations zeta-potential was determined in SA buffer, pH 6.0, for polymeric nanoparticles formed by 48 h lasting polymerization. Zeta-potentials of PANI/GOx, Ppy/GOx and PTh/GOx nanoparticles were -6.24, -9.35 and -8.64 mV, respectively.

Conclusions: 490 line the phrase “and smallest size of formed PANI/GOx, Ppy/GOx and PTh/GOx” was added and 490-492 lines were rewritten to The highest rate and smallest diameter of formed PANI/GOx, Ppy/GOx and PTh/GOx was observed in solution of sodium acetate buffer, pH 6.0.

Reviewer #1 wrote:  Materials and methods, lines121 to 123: the sentence “Polymerization bulk … polymerization course.” is not complete. A verb is missing.

Response to Reviewer #1: Materials and methods, 141-143 lines (were 121-123 lines): the verb “was used” was added in the sentence “Polymerization bulk … polymerization course.” The sentence was rewritten:Polymerization bulk solution containing 0.05 mol L−1 of glucose, 0.50 mol L−1 of polymerizable monomers (aniline, pyrrole or thiophene), 0.75 mg mL−1 of GOx was used for 48 h lasting polymerization.

Reviewer #1 wrote:  Materials and methods, lines 125 to 128: why don’t the authors study the optimization of the reaction for the thiophene monomer?

Response to Reviewer #1: The information about the optimization of polythiophene formation was added in Materials and methods:

145-148 lines (were 125-128 lines): the sentencesDuring 48 h lasting polymerization several concentrations of glucose (were 0.015, 0.02, 0.03, 0.04, 0.05, 0.06 and 0.1 mol L-1) were tested in the 0.05 mol L−1 SA buffer, pH 6.0, containing 0.75 mg mL-1 of GOx and 0.50 mol L-1 of aniline, pyrrole or thiophene.was added.

148-150 lines (were 125-128 lines): the influence of polymerization time on formation only PANI/GOx and Ppy/GOx nanoparticles was studied because PANI/GOx and Ppy/GOx composites were characterized by highest value of absorbance if compared with PTh/GOx composites. DLS and zeta-potential of PTh/GOx were investigated in present paper. The main purpose of previously investigations was the choice of more suitable polymeric nanoparticles. In our case it was PANI/GOx and Ppy/GOx nanoparticles.

Reviewer #1 wrote:  Results and discussion, lines 156 to 158: it could have been important to give a scheme showing the polymerization’s mechanism.

Response to Reviewer #1: The schema of PANI, Ppy and PTh polymerization’s mechanism was presented in Results and discussion (figure 1):

192-194 lines: the name of figure 1 was reformulated toThe formation of composite conducting polymer nanoparticles with embedded GOx by enzymatic polymerization and the schematic presentation of PANI (A), Ppy (B) and PTh (C) polymerization mechanisms.

Figure 1. The formation of composite conducting polymer nanoparticles with embedded GOx by enzymatic polymerization and the schematic presentation of PANI (A), Ppy (B) and PTh (C) polymerization mechanisms.

Results and discussion // 196-197 lines were supplemented by the phraseas strong oxidizer“.

Results and discussion // 197 line was supplemented by references18,19,…“.

Results and discussion // 195-196 lines (were 156-157 lines): the sentence “In the presence… hydrogen peroxide.” was rewritten to “In the presence of glucose and dissolved oxygen, GOx generates hydrogen peroxide and gluconolactone, which is hydrolyzed to gluconic acid in an aquous solution.

Results and discussion // 196-198 lines: the sentence Hydrogen peroxide as strong oxidizer generates radical cation of pyrrole, aniline [18,19,25] and in such way it initiates the polymerization reaction in acidic environment created by gluconic acid [27]. was corrected.

Results and discussion // lines 198-202: sentences These free radical cations undergo coupling and in this way oligomers and polymeric structures are formed [18,19]. Diffusional permeability of formed polymeric structures is sufficient, therefore, a decent substrate and product mobility towards/outwards the enzyme, which is encapsulated within formed polymer layer, is retained [27].were added.

Reviewer #1 wrote:  Results and discussion, line 165: the word “protanaited” has to be changed to “protonated”.

Response to Reviewer #1: Results and discussion // 209 line (was 165 line): the word “protanaited” was changed to “protonated”.

Reviewer #1 wrote:  Results and discussion, lines 168 to 170: the sentence “At very low pH … polymerization reaction.” has to be reformulated.

Response to Reviewer #1: Results and discussion // 168-170 lines were: the sentenceAt a very low pH value (pH < 1.0) the H+ ion concentration (in the form of H3O+ ions) increase – H+ ions are produced during the coupling of monomer [13]. was reformulated and then was deleted.

Reviewer #1 wrote:  Results and discussion // lines 183 to 186: the presence of “waves” is not obvious from the figures given by the authors.

Response to Reviewer #1: Spectras were corrected and peaks of optical absorbance are seeing well.

Fig. 1. Spectra of PANI/GOx, Ppy/GOx and PTh/GOx composite nanoparticles at pH 3.0, 6.0 and 10. (The composition of polymerization solution: 0.05 mol L−1 of glucose, 0.50 mol L−1 of aniline, pyrrole or thiophene and 0.75 mg mL−1 of glucose oxidase at pH 3.0, 6.0 and 10; 48 h lasting polymerization was performed.). 

Reviewer #1 wrote:  Results and discussion // lines 183 to 196: I am not convinced by what is written in this paragraph if I look at the figures.

Response to Reviewer #1: It is true that from figure 2 it is difficult to see the wave of absorbance for synthesized PANI/GOx, Ppy/GOx and PTh/GOx nanoparticle composites. The value of absorbance and waves were very small. However it was possible to measure the absorbance in some optical waves and, if we increased the diapazon of spectra during our measurements, we saw obviously the wave.

Results and discussion // 227-229 lines (were 183-186 lines): the sentence ”As it is seen from Figure 2A–D the UV/Vis spectra of PANI/GOx colloidal solution…” was corrected to after 48 h lasting enzymatic polymerization was characterised by one for PANI characteristic optical absorbance peak with maximum lmax = 440 nm (Figure 2A), what is in the agreement with another investigations related to synthesis of PANI [8,11,28].

Figure 2. Spectra of colloidal solution of PANI/GOx composite nanoparticles (A) and the influence of polymerization bulk solution pH on the absorbance maximum (B) at various pH values. (The composition of polymerization solution: 0.05 mol L−1 of glucose, 0.50 mol L−1 of aniline and 0.75 mg mL−1 of glucose oxidase at pH 1.0-10; 48 h lasting polymerization was performed. Optical absorbance was registered in 0.05 mol L−1 SA buffer, pH 6.0, at lmax = 440 nm.)

Reviewer #1 wrote:  Results and discussion // lines 285 to 286: the sentence “It was noticed … in aqueous solution” has to be reformulated.

Response to Reviewer #1: Results and discussion // 315-316 lines (285-286 lines): the sentenceIt should be noted that it is quite difficult to prepare solution of well-dispersed polythiophene particles because they tend to aggregate and precipitate [12,34].

Reviewer #1 wrote:  Results and discussion, lines 290 to 292: “Due to slower polymerization … nanoparticle formation.”; How can the polymerization rate and the nanoparticles formation can be improved starting form thiophene?

Response to Reviewer #1: Quite difficult to prepare well-dispersed spherical polythiophene particles without the coagulation and precipitation in low acidic solution [12,34]. To achieve stable dispersed phase are recommended to use for PTh/GOx polymerization segments of solvent-soluble and solvent-insoluble [15], the variety of combinations of initiator (cooper (II) salts, potassium peroxydisulfate) and surfactant (sodium dodecyl sulfate) [12,13]. In our research investigations were hold some conditions for polyaniline, polypyrrole and polythiophene nanoparticles with embedded glucose oxidase enzymatically synthesis. It was main reason to refuse the measurements with PTh/GOx and concentrated at PANI/GOx and Ppy/GOx composite nanoparticles formation.

Reviewer #1 wrote:  Results and discussion, line 308: the word “lasing” has to be changed to “lasting”, I guess.

Response to Reviewer #1: Results and discussion // 338 line (was 308 line): the word “lasing” was changed to “lasting”.

Reviewer #1 wrote:  Results and discussion, line 341: the word “pats” has to be changed to “part”, I think.

Response to Reviewer #1: Results and discussion // 341 line was: the word “pats” was changed to “parts” and then deleted.

Reviewer #1 wrote:  Results and discussion, line 358: the word “oxidaized” has to be changed to “oxidized”.

Response to Reviewer #1: Results and discussion // 419 line (was 358 line): the word “oxidaized” was changed to “oxidized”.

Reviewer #1 wrote:  Conclusions, line 410: I don’t agree with the authors when they wrote that they synthesized PTh/Gox-based nanoparticles. Indeed, they just described the procedure they used but concluded that the polymerization rate was too slow to be able to observe the formation of any nanoparticles or aggregates. Therefore, the authors have either to improve their synthesis starting form thiophene and given detailed characteristics of the corresponding materials or erase both in their title and their manuscript the PTh/Gox-based materials.

During our investigations nanoparticles of PTh/GOx were synthesized. The optimal solution of enzymatic polymerization, the diameter, zeta-potential of formed PTh/GOx particles by DLS were investigated. The further investigations (polymerization time) were performed for PANI/GOx and Ppy/GOx particles because these kind of composites were characterized by highest value of absorbance and smaller nanoparticles.

Conclusions: the information about PTh/GOx nanoparticles formation was added:

486-488 lines (411-413 lines were): the sentence “PANI/GOx and Ppy/GOx nanoparticles were characterized … respectively.” was rewritten toPANI/GOx, Ppy/GOx and PTh/GOx nanoparticles were characterized by optical absorbance maximums at lmax = 440 nm, lmax = 460 nm and lmax = 450 nm, respectively.

490-492 lines (415-416 lines were): the sentence “The highest rate … pH 6.0.” was rewritten toThe highest rate and smallest diameter of formed PANI/GOx, Ppy/GOx and PTh/GOx was observed in solution of sodium acetate buffer, pH 6.0.

Another corrections:

Abstract, line 29 (was  line 25): the word “96 h” was changed to in the range of 48-96 h.

Keywords, line 33: the word “polymeric” was deleted in the phrase “polymeric nanoparticles”.

Introduction, 37 line (was line 32): the space was deleted and the phrase such as in the phrase “…(CPs) such as polyaniline…” was supplemented.

Introduction, 40 line: the coma was deleted and the word and in the phrase ”… nanotoxic, stable…”. The phrase was rewritten as “…nanotoxic and stable …”

Introduction, 47-48 lines (were 42-43 lines): the sentence “PANI, Ppy and PTh can be oxidatively or reductively doped by corresponding ions dependently on the pH value of polymerization bulk solution [1,5,6,18,19,20].” was rewritten to PANI, Ppy and PTh can be doped by corresponding ions dependently on the pH value of polymerization bulk solution during oxidation or reduction processes [1,5,6,18,19,20].

Introduction, 49-50 lines (were 43-44 lines): the sentence “A novel strategies … the last decade.” was rewritten to Novel strategies for the enzymatic synthesis of PANI [21,22], Ppy [23,24,25,26] and PTh [27] were developed during the last decade.

Introduction, 57 line (was 52 line): the word “depends” was changed to “depend”.

Introduction, 72-73 lines (were 64-65 lines): the phrase “Soluble in aqueous solution Ppy can be synthesized by various methods and used..” was changed toPpy in aqueous solution can be synthesized by various methods and can be used…”

Introduction, 83 line was: the word “if” in the phrase “if polymerization” was deleted.

In all text the word “until” was changed to “to”.

Materials and methods, 159 line: the name of the part2.5. The Evaluation of PANI/GOx and Ppy/GOx Nanoparticle Formation by DLSwas changed to 2.5. The evaluation of PANI/GOx, Ppy/GOx and PTh/GOx composite nanoparticles formation by DLS

Materials and methods, 168-170 lines (were 142-144 lines): the sentence “Zetasizer Nano ZS from Malvern (Herrenberg, Germany) equipped with a 633 nm He-Ne laser and 142 operating at 173° angle of DLS was used to evaluate the diameter of PANI/GOx and Ppy/GOx 143 particles.” was changed to Diameter and zeta-potential of PANI/GOx, Ppy/GOx and PTh/GOx composite nanoparticles were evaluated by DLS and microelectrophoresis, using the Zetasizer Nano ZS from Malvern (Herrenberg, Germany) equipped with a 633 nm He-Ne laser.

Results and discussion, 204 line and References: the publication 39 and 39 German, N.; Ramanaviciene, A.; Ramanavicius, A. Formation of polyaniline and polypyrrole nanocomposites with embedded glucose oxidase and gold nanoparticles. Polymers 2019, 11, 377 (13 pp.). DOI:10.3390/polym11020377was added in previously paper.

Results and discussion, 260 line (was 226 line): the word “were ” was changed to has been.

Results and discussion, 261 line (was 226 line): the phraseand were in an agreement” was changed to , what is in agreement.

Results and discussion, 278 line (was 247 line): after phrase “In all cases” the comma was added In all cases,”.

Results and discussion, 288 line (was 259 line): the word “were” was changed to has been”.

Results and discussion, 334-336 lines (were 292-294 lines): the sentence “Therefore, further investigations were focused on the enzymatic formation of PANI/GOx and Ppy/GOx composite nanoparticles.” was supplemented by the phraseof a polymerization’s durationand was rewritten by Therefore, further investigations of a polymerization’s duration were focused on the enzymatic formation of PANI/GOx and Ppy/GOx composite nanoparticles.

Results and discussion, 348 line: the publication 39 was added in the phrasecatalysed reaction rate and the highest stability [24,39]”.

Results and discussion, 413 line: the number of the part3.3. The influence of polymerization duration on PANI/GOx and Ppy/GOx composite nanoparticles formation” was changed to3.4…….”.

Results and discussion, 437 line (was 369 line): the word “10 days” was changed to 168 h”.

Results and discussion, 442 line (was 372 line): the word “96 h” was changed to 48 h”.

Results and discussion, 444 line (was 373 line): the word “96 h” was changed to 48-96 h”.

Results and discussion, 461 line (was 361 line): the word “high” in the phrase “…characterized by high diameter were formed… was changed tolargeand it was changed to “…characterized by large diameter were formed…”.

Results and discussion: the number of figure “6“ was changed to 8”.

Results and discussion: the number of figure “7“ was changed to 9”.

Results and discussion, 478-479 lines (was 408 line): the phrase “, SD – standard deviation.” was added after  “… Figure 9” and before “)”.

Results and discussion, 454, 477 lines (were 383, 407 lines): the number of Table “1“ was changed to 2”.

Conclusions, 504 line (426 line was): the citation 39 was supplemented to the phrase “…have been demonstrated [37,39]…”.

The parts of “Authors Contributions” and “Conflict of Interest” (511-515 lines) were supplemented to the publication:

Author Contributions: conceptualization, Ar.R. and N.G.; methodology, N.G.; software, N.G.; validation, A.R., Ar.R. and N.G.; formal analysis, N.G.; investigation, N.G. and A.P.; resources, Ar.R..; data curation, A.R.; writing-original draft preparation, N.G.; writing-review and editing, Ar.R.; visualization, A.R.; supervision, Ar.R.; project administration, Ar.R.; funding acquisition, Ar.R.

Conflict of Interest: The authors declare no conflict of interest.

The part of “References” was corrected according the requirements of journal Nanomaterials. 

We will thank for positive feedback and valuable recommendations.

We hope after all these corrections our manuscript is suitable for publication.

Yours sincerely,
Arunas Ramanavicius

----------------------------------------------------------------
Prof. habil. dr. Arunas Ramanavicius

Head of Department of Physical Chemistry,

Faculty of Chemistry, Vilnius University,

Naugarduko 24, 03225 Vilnius 6, Lithuania; e-mail: [email protected]

Reviewer 2 Report

I agree with the changes and amendments made. Two minor comments:

Table 1 added: it is suggested to present the results expressed in round numbers (180+/-94 as an example), 

Line 482: for the same reason, please lease one decimal for zeta potentials (-6.2 instead of -6.24 etc.)

Author Response

(The authors gave the same response as above.)

Reviewer 3 Report

The manuscript has been revised to provide more information. Publish it as current form.

Author Response

Response to reviewer #3:

We would like to thank the reviewer for very professional review of our manuscript, valuable comments and recommendations. Thank you for pointing out our mistakes and giving suggestions which further on improve clarity of this paper. We did our best in order to improve the manuscript according to comments and recommendations. All the most important changes are highlighted in the revised manuscript. Corrections and changes are highlighted in the manuscript (in red).

Please   find below short explanations and answers to your questions during 1st    round of evaluation, additional   corrections after second round are indicated in red:

Reviewer #3 wrote:   PANI, Ppy and PTh nanoparticles with embedded GOx were synthesized at room temperature in darkness using polymerization bulk solution. What kind of polymerization was exploited anyway? Please provide the NMR and GPC data to confirm the structure.

PANI, Ppy and PTh nanoparticles with embedded GOx were synthesized at room temperature in darkness using enzymatic polymerization, because only enzyme was added in the bulk polymerization solution. Any another oxidative compounds and applied electrochemical potential weren’t used during this kind of polymeric compounds formation.

We are very sorry, but it isn’t possible to provide the NMR and GPC data to confirm the structure of formed polymeric particles. We haven’t any equipment for gel permeation chromatography - nuclear magnetic resonance (GPC-NMR) analysis in our institution. We also haven’t any resources to pay for this kind of analysis in another institutions.

Reviewer #3 wrote:   Why the GOx could be embedded in the polymers during the polymerization?

Response to Reviewer #3: Enzyme – glucose oxidase is considered effective catalyst, which, in the presence of glucose is able to produce hydrogen peroxide. Hydrogen peroxide as an oxidizer generates free radicals [18,19,Error! Bookmark not defined.] and in such way it initiates the polymerization reaction in low acidic environment created by hydrolyzed lactone of gluconic acid [Error! Bookmark not defined.]. Owing to the diffusional permeability of the polymeric nanoparticles, a decent substrate and product mobility towards/outwards the enzyme is retained [27].

The information about polymers/GOx polymerization was added in Results and discussion: 195-198 lines (156-158 lines were): sentences “In the presence of glucose … of gluconic acid [27].“ were rewritten toIn the presence of glucose and dissolved oxygen, GOx generates hydrogen peroxide and gluconolactone, which is hydrolyzed to gluconic acid in an aquous solution. Hydrogen peroxide as strong oxidizer generates radical cation of pyrrole, aniline [18,19,Error! Bookmark not defined.] and in such way it initiates the polymerization reaction in acidic environment created by gluconic acid [Error! Bookmark not defined.].

198-202 lines: sentences were addedThese free radical cations undergo coupling and in this way oligomers and polymeric structures are formed [18,19]. Diffusional permeability of formed polymeric structures is sufficient, therefore, a decent substrate and product mobility towards/outwards the enzyme, which is encapsulated within formed polymer layer, is retained [27].

Enzyme molecules are embedded inside of polymeric nanoparticles. It is mean that polyaniline, polypyrrole and polythiophene are able to make the polymeric layer abound glucose oxidase. This effect was investigated in the previously papers, where graphite rod electrodes were firstly immobilized by GOx and only then modified by Ppy [23] and PTh [27] layers. The embedding of enzymes within polymeric film prevents the enzyme from being leached out, while at the same time maintains the accessibility of the catalytic sites due to the permeability of the film to analytes [Pub. 1]. Polymer films on electrode surfaces enabled to increase the concentration of entrapped enzyme by thin layers of polypyrrole derivatives [Pub. 2].

[Pub. 1] Ho, P.K.H.; Kim, J.S.; Burroughes, J.H.; Becker, H.; Li, S.F.Y.; Brown, T.M.; Cacialli, F.; Friend, R.H. Molecular-scale interface engineering for polymer light-emitting diodes. Nature 2000, 404, 481-483.

[Pub 2] Schuhmann, W.; Kranz, C.; Wohlschlager, H.; Strohmeier, J. Pulse technique for the electrochemical deposition of polymer films on electrode surfaces. Biosens. Bioelectron. 1997, 12, 1157 -1167.

Reviewer #3 wrote:   Please provide the SEM and TEM images of PANI, Ppy and PTh nanoparticles with embedded GOx.

Response to Reviewer #3: The morphology of PANI/GOx and Ppy/GOx was evaluated by emission scanning electron microscopy (FE-SEM) and described in this paper.

Materials and methods, 173-182 lines: the paragraph was supplemented:

2.6. Imaging of PANI/GOx and Ppy/GOx composite nanoparticles by FE-SEM

The enzymatic polymerization of PANI/GOx and Ppy/GOx composite nanoparticles was proceeded in darkness at room temperature within 24 h. The procedure of the formation, separation, washing and collection of nanoparticles was performed similarly as it was described in chapter 2.2. To reduce the effect of dissolved salts colloidal solutions of composite nanoparticles were diluted in 50 l of deionized water and homogenized using an ultrasound Bandelin Sonorex RK 31 H (Berlin, Germany). Then, 6 l of nanoparticle solution was placed on surface of graphite rod and dried, this procedure was repeated for 8 times and then the surface was characterized by Hitachi SU-70 field emission scanning microscope equipped by EDS and EBSD modules from Hitachi (Dublin, Irland). Turbosputer was applied for preparation of synthesized CPs for SEM-based imaging.

Results and discussion: 370-371 lines were supplemented: 3.3. The evaluation of PANI/GOx, Ppy/GOx and PTh/GOx composite nanoparticles formation by DLS and the morphology of PANI/GOx, Ppy/GOx nanocomposites”.

Results and discussion, 398-412 lines: paragraphs about DLS investigations were supplemented:

Some researchers evaluated the diameter of synthesized polymer nanocomposites by FE-SEM. During the enzymatic polymerization of aniline using chitosan and poly(N-isopropylacrylamide) as steric stabilizers 50 nm PANI nanoparticles were synthesized [Error! Bookmark not defined.]. In our study PANI/GOx and Ppy/GOx composite nanoparticles formed during 24 h lasting enzymatic polymerization were investigated by FE-SEM (Figure 7).                    Figure 7. FE-SEM images of PANI/GOx (A) and Ppy/GOx (B) composite nanoparticles after 24 h lasting enzymatic polymerization. (The polymerization solution: 0.05 mol L-1 SA buffer, pH 6.0, with 0.05 mol L-1 of glucose, 0.75 mg mL-1 of GOx and 0.50 mol L-1 of aniline or pyrrole.)

During enzymatic polymerization spherical PANI/GOx and Ppy/GOx composite nanoparticles were formed. It is seen that PANI/GOx (Figure 7A) nanoparticles were randomly distributed on the surface of graphite rod used for investigations. Ppy/GOx particles were densely sti cked together (Figure 7B). The diameter of nanoparticles depends on the kind of polymerized monomer. It was determined that diameter of PANI/GOx and Ppy/GOx composites was in the range of 38-50 and 50-100 nm, respectively.

Conclusions: 493-494 lines were supplemented by a sentence Spherical particles of PANI/GOx and Ppy/GOx were formed during enzymatic polymerization.”.

Reviewer #3 wrote:   Please provide the DLS and zeta-potential data of PANI, Ppy and PTh nanoparticles with embedded GOx in various pH solution to confirm the pH effect.

Response to Reviewer #3: DLS and zeta-potential investigations to confirm the pH effect during PANI/GOx, Ppy/GOx and PTh/GOx formation were performed and presented in this paper:

Introduction, 109-110 lines (89-90 lines were): the information about PTh/GOx nanoparticles formation was added and the sentence was rewrittenThe diameter of PANI/GOx, Ppy/GOx and PTh/GOx nanoparticles was evaluated using dynamic light scattering (DLS) technique.

Materials and methods: 159-172 lines were supplemented by the information about DLS and zeta-potential for PANI/GOx, Ppy/GOx and PTh/GOx:

2.5. The evaluation of PANI/GOx, Ppy/GOx and PTh/GOx composite nanoparticles formation by DLS

The influence of solution’s pH on the diameter and zeta-potential of PANI/GOx, Ppy/GOx and PTh/GOx composite nanoparticles was studied in pH range from 2.0 to 8.0 at +20 ± 2 °C. In this experiment 0.01 or 0.001 mol L1 HCl, 0.05 mol L1 SA buffer, pH 6.0, and 1 × 106 mol L1 NaOH were used to prepare solutions in pH range between 2.0 and 8.0. Polymerization solutions containing 0.05 mol L1 of glucose, 0.50 mol L1 of aniline, pyrrole or thiophene and 0.75 mg mL1 of GOx were monitored during 15-120 h lasting polymerization.

Polymeric nanoparticles after enzymatic synthesis were separated, washed two times with 0.05 mol L−1 SA buffer, pH 6.0, and collected by the centrifugation (6 min, 16.1 × 103 g) as it is described in chapter 2.2. Diameter and zeta-potential of PANI/GOx, Ppy/GOx and PTh/GOx composite nanoparticles were evaluated by DLS and microelectrophoresis, using the Zetasizer Nano ZS from Malvern (Herrenberg, Germany) equipped with a 633 nm He-Ne laser. The obtained data was analysed with Dispersion Technology Software version 6.01 from Malvern. DLS and microelectrophoresis investigations were evaluated and visualized by SigmaPlot software 12.5.

Results and discussion: 370-371 lines were supplemented 3.3. The evaluation of PANI/GOx, Ppy/GOx and PTh/GOx composite nanoparticles formation by DLS and the morphology of PANI/GOx, Ppy/GOx nanocomposites”.

Results and discussion, 372-397 lines: paragraphs of investigations by DLS were supplemented:

Depending on pH the negatively or positively charged enzymes can be entrapped within the CPs during oxidation/reduction reactions [Error! Bookmark not defined.]. Enzymatic polymerization is strongly depending on solutions pH. For instance, the conducting form of linear PANI is synthesized at pH 4.0-4.5 , wherein at pH 6.0 or higher – more branched and insulating form of PANI is formed [Error! Bookmark not defined.,Error! Bookmark not defined.].

The diameter of 48 h enzymatically synthesized PANI-, Ppy- and PTh-based nanocomposites with embedded GOx in solution of pH 2.0-8.0 was evaluated by DLS technique. Results of PANI/GOx, Ppy/GOx and PTh/GOx composite nanoparticles diameter and their distribution are presented in Table 1 and Figure 6, respectively. The diameter of PANI/GOx, Ppy/GOx and PTh/GOx composites nanoparticles depended on pH of polymerization solution and it decreased by the increase of pH. It was determined that in SA buffer, pH 6.0, PANI/GOx, Ppy/GOx and PTh/GOx composite nanoparticles of 53.9, 85.8 and 98.2 nm diameter were formed, respectively. DLS results (Table 1) represent that formed PANI/GOx composite nanoparticles were by 1.59 and 1.82 times smaller if compared with Ppy/GOx and PTh/GOx. Due to relatively large (535 nm) diameter of PTh/GOx, which were formed at pH 8.0, we decided that such relatively large particle colloidal solution will be not very stable and focussed mostly on the investigation of PANI/GOx and Ppy/GOx composite nanoparticles, which are smaller, therefore it was expected that colloidal solution of these nanoparticles will be more stable.

Table 1. The dependence of PANI/GOx, Ppy/GOx and PTh/GOx composite nanoparticles diameter on the pH of solutions. The composition of polymerization solution: 0.05 mol L-1 of glucose, 0.75 mg mL-1 of GOx and 0.50 mol L-1 of aniline, pyrrole or thiophene. Particles were formed by 48 h lasting enzymatic polymerization. The diameter of particles was determined by DLS in 0.05 mol L-1 SA buffer, pH 6.0.

pH

Diameter, nm

PANI/GOx

Ppy/GOx

PTh/GOx

2.0

180 ±94.3

180 ±45.5

   169 ±61.6

5.0

56.1 ±21.8

155 ±61.0

62.5 ±9.36

6.0

53.9 ±6.86

85.8 ±17.7

98.2 ±28.0

8.0

106 ±41.3

308 ±48.5

535 ±106

Figure 6. The distribution of PANI/GOx, Ppy/GOx and PTh/GOx composite nanoparticles diameter. Particles were formed by 48 h lasting enzymatic polymerization at pH 6.0.

Results and discussion, 480-483 lines: paragraph of zeta-potential investigations was supplemented:

The electrical charge of particles is usually expressed by zeta-potential. In our investigations zeta-potential was determined in SA buffer, pH 6.0, for polymeric nanoparticles formed by 48 h lasting polymerization. Zeta-potentials of PANI/GOx, Ppy/GOx and PTh/GOx nanoparticles were -6.24, -9.35 and -8.64 mV, respectively.

Conclusions: 490 line the phrase “and smallest size of formed PANI/GOx, Ppy/GOx and PTh/GOx” was added and 490-492 lines were rewritten to The highest rate and smallest diameter of formed PANI/GOx, Ppy/GOx and PTh/GOx was observed in solution of sodium acetate buffer, pH 6.0.

Reviewer #3 wrote:   Absorptance of PANI, Ppy and PTh nanoparticles without embedded GOx should be provided for comparison.

Response to Reviewer #3: The absorbance of PANI, Ppy and PTh nanoparticles without embedded glucose oxidase was investigated in our measurements. The comparison of absorbance for PANI, Ppy and PTh was added in description of results and in figure 5A:

Results and discussion: phrases were added in paragraphs:

333 line the phrase , Ppy and PTh”,

335ine the phrase , pyrrole or thiophene”,

335-336 lines the phrase “Any autopolymerization was observed in the pH range from 1.0 to 12 for thiophene. The…”,

338 line the phrase 0.035 … lasting”,

340 line the phrase “…1.20…”,

and 332-340 lines (304-310 lines were), sentences “To evaluate the influence … at pH 2.0.” were reformulated as To evaluate the influence of the autopolymerization on the formation of PANI, Ppy and PTh an experiment was performed in solutions containing only 0.05 mol L−1 of glucose and 0.5 mol L−1 of aniline, pyrrole or thiophene at pH from 1.0 to 12 without any GOx. Any autopolymerization was observed in the pH range from 1.0 to 12 for thiophene. The slow autopolymerization has been observed only at pH 1.0 for aniline and at pH below 2.0 for pyrrole and it reached 0.030 and 0.035 a.u. after 48 h lasting polymerization, respectively. At higher pH value this autopolymerization phenomenon was not observed. At pH 1.0 in the absence of GOx the polymerization rate of aniline was 1.20 times slower in the comparison of pyrrole at pH 2.0.” 

Results and discussion, figure 5A: the autopolymerization’s diagrams for PANI and Ppy were added in figure 5 (columns of grey colour with lines). The description and conditions of figure 5 (319-330 lines (295-301 lines were)) were reformulated:

Figure 5. The diagram of optical absorbance of PANI-, Ppy- and PTh-based composite nanoparticles in the presence and absence of GOx (A) … . (A: The composition of polymerization solution, pH 6.0: 0.05 mol L−1 of glucose, 0.50 mol L−1 of aniline, pyrrole or thiophene and 0.75 mg mL−1 of glucose oxidase (columns of grey colour); the composition of solution used for autopolymerization: 0.05 mol L−1 of glucose, 0.50 mol L−1 of aniline or pyrrole at pH 1.0 or pH 2.0, respectively (columns of grey colour with lines); … . Polymerization lasted for 48 h.). 

Another corrections:

Abstract, line 29 (was  line 25): the word “96 h” was changed to in the range of 48-96 h.

Keywords, line 33: the word “polymeric” was deleted in the phrase “polymeric nanoparticles”.

Introduction, 37 line (was line 32): the space was deleted and the phrase such as in the phrase “…(CPs) such as polyaniline…” was supplemented.

Introduction, 40 line: the coma was deleted and the word and in the phrase ”… nanotoxic, stable…”. The phrase was rewritten as “…nanotoxic and stable …”

Introduction, 47-48 lines (were 42-43 lines): the sentence “PANI, Ppy and PTh can be oxidatively or reductively doped by corresponding ions dependently on the pH value of polymerization bulk solution [1,5,6,18,19,20].” was rewritten to PANI, Ppy and PTh can be doped by corresponding ions dependently on the pH value of polymerization bulk solution during oxidation or reduction processes [1,5,6,18,19,20].

Introduction, 49-50 lines (were 43-44 lines): the sentence “A novel strategies … the last decade.” was rewritten to Novel strategies for the enzymatic synthesis of PANI [21,22], Ppy [23,24,25,26] and PTh [27] were developed during the last decade.

Introduction, 57 line (was 52 line): the word “depends” was changed to “depend”.

Introduction, 72-73 lines (were 64-65 lines): the phrase “Soluble in aqueous solution Ppy can be synthesized by various methods and used..” was changed toPpy in aqueous solution can be synthesized by various methods and can be used…”

Introduction, 83 line was: the word “if” in the phrase “if polymerization” was deleted.

In all text the word “until” was changed to “to”.

Materials and methods, 159 line: the name of the part2.5. The Evaluation of PANI/GOx and Ppy/GOx Nanoparticle Formation by DLSwas changed to 2.5. The evaluation of PANI/GOx, Ppy/GOx and PTh/GOx composite nanoparticles formation by DLS

Materials and methods, 168-170 lines (were 142-144 lines): the sentence “Zetasizer Nano ZS from Malvern (Herrenberg, Germany) equipped with a 633 nm He-Ne laser and 142 operating at 173° angle of DLS was used to evaluate the diameter of PANI/GOx and Ppy/GOx 143 particles.” was changed to Diameter and zeta-potential of PANI/GOx, Ppy/GOx and PTh/GOx composite nanoparticles were evaluated by DLS and microelectrophoresis, using the Zetasizer Nano ZS from Malvern (Herrenberg, Germany) equipped with a 633 nm He-Ne laser.

Results and discussion, 204 line and References: the publication 39 and 39 German, N.; Ramanaviciene, A.; Ramanavicius, A. Formation of polyaniline and polypyrrole nanocomposites with embedded glucose oxidase and gold nanoparticles. Polymers 2019, 11, 377 (13 pp.). DOI:10.3390/polym11020377was added in previously paper.

Results and discussion, 260 line (was 226 line): the word “were ” was changed to has been.

Results and discussion, 261 line (was 226 line): the phraseand were in an agreement” was changed to , what is in agreement.

Results and discussion, 278 line (was 247 line): after phrase “In all cases” the comma was added In all cases,”.

Results and discussion, 288 line (was 259 line): the word “were” was changed to has been”.

Results and discussion, 334-336 lines (were 292-294 lines): the sentence “Therefore, further investigations were focused on the enzymatic formation of PANI/GOx and Ppy/GOx composite nanoparticles.” was supplemented by the phraseof a polymerization’s durationand was rewritten by Therefore, further investigations of a polymerization’s duration were focused on the enzymatic formation of PANI/GOx and Ppy/GOx composite nanoparticles.

Results and discussion, 348 line: the publication 39 was added in the phrasecatalysed reaction rate and the highest stability [Error! Bookmark not defined.,Error! Bookmark not defined.]”.

Results and discussion, 413 line: the number of the part3.3. The influence of polymerization duration on PANI/GOx and Ppy/GOx composite nanoparticles formation” was changed to3.4…….”.

Results and discussion, 437 line (was 369 line): the word “10 days” was changed to 168 h”.

Results and discussion, 442 line (was 372 line): the word “96 h” was changed to 48 h”.

Results and discussion, 444 line (was 373 line): the word “96 h” was changed to 48-96 h”.

Results and discussion, 461 line (was 361 line): the word “high” in the phrase “…characterized by high diameter were formed… was changed tolargeand it was changed to “…characterized by large diameter were formed…”.

Results and discussion: the number of figure “6“ was changed to 8”.

Results and discussion: the number of figure “7“ was changed to 9”.

Results and discussion, 478-479 lines (was 408 line): the phrase “, SD – standard deviation.” was added after  “… Figure 9” and before “)”.

Results and discussion, 454, 477 lines (were 383, 407 lines): the number of Table “1“ was changed to 2”.

Conclusions, 504 line (was 426 line): the citation 39 was supplemented to the phrase “…have been demonstrated [37,39]…”.

The parts of “Authors Contributions” and “Conflict of Interest” (511-515 lines) were supplemented to the publication:

Author Contributions: conceptualization, Ar.R. and N.G.; methodology, N.G.; software, N.G.; validation, A.R., Ar.R. and N.G.; formal analysis, N.G.; investigation, N.G. and A.P.; resources, Ar.R..; data curation, A.R.; writing-original draft preparation, N.G.; writing-review and editing, Ar.R.; visualization, A.R.; supervision, Ar.R.; project administration, Ar.R.; funding acquisition, Ar.R.

Conflict of Interest: The authors declare no conflict of interest.

The part of “References” was corrected according the requirements of journal Nanomaterials. 

We will thank for positive feedback and valuable recommendations.

We hope after all these corrections our manuscript is suitable for publication.

Yours sincerely,
Arunas Ramanavicius

----------------------------------------------------------------
Prof. habil. dr. Arunas Ramanavicius

Head of Department of Physical Chemistry,

Faculty of Chemistry, Vilnius University,

Naugarduko 24, 03225 Vilnius 6, Lithuania; e-mail: [email protected]

This manuscript is a resubmission of an earlier submission. The following is a list of the peer review reports and author responses from that submission.

Round  1

Reviewer 1 Report

In their manuscript, German et al. studied the enzymatic synthesis of poly(aniline), poly(pyrolle) and poly(thiophene) based nanoparticles using glucose oxidase. Despites the interest of the results presented in this manuscript, both the language and the punctuation have to be carefully checked and corrected in order to improve the overall style of the manuscript and render it easier to read and understand. Moreover, the figures 2 to 4 and 6, more specifically the UV spectra, need to be improved: they are too small and the graduation of the X axis needs to be given with more details (the same as the one described in the text). Finally, the authors said that they will present the enzymatic synthesis of poly(thiophene) based nanoparticles embedded glucose oxidase, but the just said that the polymerization rate was to slow to observe the formation of any nanoparticles without further studies or perspectives.

In view of these general comments and those given below, I do recommend this manuscript for publication in Nanomaterials after major revision.

Please find below, specific comments on the manuscript:

1. Abstract: what about the results obtained for the PTh-based nanoparticles? The authors have to add few words on these formulations: they mentioned them in the title and the beginning of the abstract.

2. Introduction, line 34: the word ”, respectively” has to be added at the end of the sentence “The electrical conductivity … S cm-1” just before the references.

3. Introduction, line 36: the word “stabile” has to be changed to “stable”.

4. Introduction, line 43: “A novel strategies” has to be changed to “Novel strategies”

5. Introduction, lines 55 to 61: these sentences are not clear and have to be reformulated. Moreover, what happen when the pH is in the range of 2 and 4.6? Do the authors think it can be possible to apply such low pH for an enzymatic reaction? Don’t they think that the enzyme will be inactivated at pH 1?

6. Introduction, lines 89 to 90: Same remark as the 1st one, what about PTh/Gox materials? Don’t they form nanoparticles? In the experimental part, the authors described the formation of Pth/Gox based nanoparticles?

7. Materials and methods, lines121 to 123: the sentence “Polymerization bulk … polymerization course.” is not complete. A verb is missing.

8. Materials and methods, lines 125 to 128: why don’t the authors study the optimization of the reaction for the thiophene monomer?

8. Results and discussion, lines 156 to 158: it could have been important to give a scheme showing the polymerization’s mechanism.

9. Results and discussion, line 165: the word “protanaited” has to be changed to “protonated”.

10. Results and discussion, lines 168 to 170: the sentence “At very low pH … polymerization reaction.” has to be reformulated.

11. Results and discussion, lines 183 to 186: the presence of “waves” is not obvious from the figures given by the authors.

12. Results and discussion, lines 183 to 196: I am not convinced by what is written in this paragraph if I look at the figures.

13. Results and discussion, lines 285 to 286: the sentence “It was noticed … in aqueous solution” has to be reformulated.

14. Results and discussion, lines 290 to 292: “Due to slower polymerization … nanoparticle formation.”; How can the polymerization rate and the nanoparticles formation can be improved starting form thiophene?

15. Results and discussion, line 308: the word “lasing” has to be changed to “lasting”, I guess.

16. Results and discussion, line 341: the word “pats” has to be changed to “part”, I think.

17. Results and discussion, line 358: the word “oxidaized” has to be changed to “oxidized”.

18. Conclusions, line 410: I don’t agree with the authors when they wrote that they synthesized PTh/Gox-based nanoparticles. Indeed, they just described the procedure they used but concluded that the polymerization rate was too slow to be able to observe the formation of any nanoparticles or aggregates. Therefore, the authors have either to improve their synthesis starting form thiophene and given detailed characteristics of the corresponding materials or erase both in their title and their manuscript the PTh/Gox-based materials.

Reviewer 2 Report

The authors have performed enzyme based polymerization of aniline, pyrrole and thiophene and investigated size distribution and optical characteristics of the products. The manuscript can be recommended for publication after the following changes and amendments:

1.     The choice of glucose oxidase as polymerization trigger is not explained.

2.     Long reaction time does not give any doubts in partial inactivation of the enzyme especially in extreme pH region. This is not taken into account in most discussion

3.     Heterogeneous character of the reaction media complicates the use of UV-vis spectroscopy. Full spectra are required at least for a number of reaction mixture instead of pictograms presented.

4.     DLS offers much more opportunities including the determination of zeta potential and histograms of size distribution. This should be used for explanation of the behavior of reaction systems considered.

5.     Introduction should be extended by description of direct polymerization of aniline used as substrate in peroxidase aided reaction.

6.     The results are rather far from biosensor applications because electroconductivity of the products was neither characterized nor used for enzyme wiring. It is even not clear where enzyme molecules are located, inside the particles or in solution.

7.     The role of enzyme reaction could be considered by variation of enzyme : substrate ratio.

Reviewer 3 Report

Comment

The present manuscript reports Enzymatic Formation of Polyaniline, Polypyrrole and Polythiophene Nanoparticles with Embedded 3 Glucose Oxidase. Discussions on these points are informative to readers.

1. PANI, Ppy and PTh nanoparticles with embedded GOx were synthesized at room temperature in darkness using polymerization bulk solution. What kind of polymerization was exploited anyway? Please provide the NMR and GPC data to confirm the structure.

2. Why the GOx could be embedded in the polymers during the polymerization?

3. Please provide the SEM and TEM images of PANI, Ppy and PTh nanoparticles with embedded GOx.

4. Please provide the DLS and zeta-potential data of PANI, Ppy and PTh nanoparticles with embedded GOx in various pH solution to confirm the pH effect.

5. Absorptance of PANI, Ppy and PTh nanoparticles without embedded GOx should be provided for comparison.